# NEARLY LOSSLESS ADAPTIVE BIT SWITCHING

## ABSTRACT

Model quantization is widely applied for compressing and accelerating deep neural networks (DNNs). However, conventional Quantization-Aware Training (QAT) focuses on training DNNs with uniform bit-width. The bit-width settings vary across different hardware and transmission demands, which induces considerable training and storage costs. Hence, the scheme of one-shot joint training multiple precisions is proposed to address this issue. Previous works either store a larger FP32 model to switch between different precision models for higher accuracy or store a smaller INT8 model but compromise accuracy due to using shared quantization parameters. In this paper, we introduce the ***Double Rounding*** quantization method, which fully utilizes the quantized representation range to accomplish nearly lossless bit-switching while reducing storage by using the highest integer precision instead of full precision. Furthermore, we observe a competitive interference among different precisions during one-shot joint training, primarily due to inconsistent gradients of quantization scales during backward propagation. To tackle this problem, we propose an Adaptive Learning Rate Scaling (**ALRS**) technique that dynamically adapts learning rates for various precisions to optimize the training process. Additionally, we extend our *Double Rounding* to one-shot mixed precision training and develop a Hessian-Aware Stochastic Bit-switching (**HASB**) strategy. Experimental results on the ImageNet-1K classification demonstrate that our methods have enough advantages to state-of-the-art one-shot joint QAT in both multi-precision and mixed-precision. We also validate the feasibility of our method on detection and segmentation tasks, as well as on LLMs. Our codes are available at `https://anonymous.4open.science/r/Double-Rounding-EF78/README.md`.

## 1 INTRODUCTION

Recently, with the popularity of mobile and edge devices, more and more researchers have attracted attention to model compression due to the limitation of computing resources and storage. Model quantization (Zhou et al., 2016; Esser et al., 2019) has gained significant prominence in the industry. Quantization maps floating-point values to integer values, significantly reducing storage requirements and computational resources without altering the network architecture.

Generally, for a given pre-trained model, the quantization bit-width configuration is predefined for a specific application scenario. The quantized model then undergoes retraining, *i.e.*, QAT, to mitigate the accuracy decline. However, when the model is deployed across diverse scenarios with different precisions, it often requires repetitive retraining processes for the same model. A lot of computing resources and training costs are wasted. To address this challenge, involving the simultaneous training of multi-precision (Jin et al., 2020; Xu et al., 2022) or one-shot mixed-precision (Jin et al., 2020; Xu et al., 2023) have been proposed. Among these approaches, some involve sharing weight parameters between low-precision and high-precision models, enabling dynamic bit-width switching during inference.

However, bit-switching from high precision (or bit-width) to low precision may introduce significant accuracy degradation due to the *Rounding* operation in the quantization process. Additionally, there is severe competition in the convergence process between higher and lower precisions in multi-precision scheme. In mixed-precision scheme, previous methods often incur vast searching and retraining costs due to decoupling the training and search stages. Due to the above challenges, bit-switching remains a very challenging problem. Our motivation is designing a bit-switching

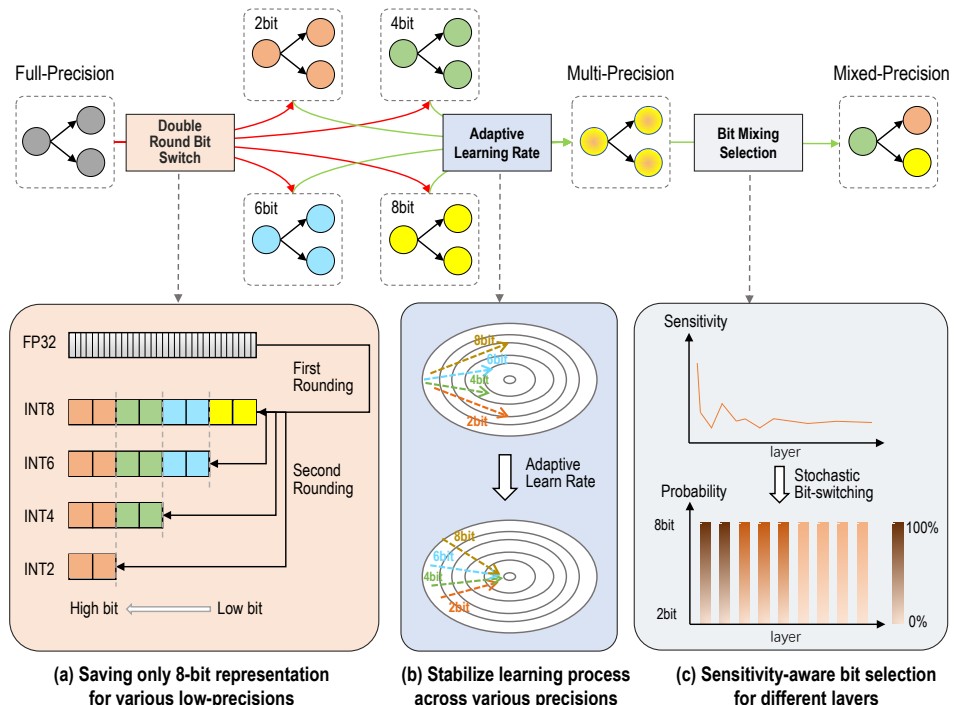

Figure 1: Overview of our proposed lossless adaptive bit-switching strategy.

quantization method that doesn't require storing a full-precision model and achieves nearly lossless switching from high-bits to low-bits. Specifically, for different precisions, we propose unified representation, normalized learning steps, and tuned probability distribution so that an efficient and stable learning process is achieved across multiple and mixed precisions, as depicted in Figure 1.

To solve the bit-switching problem, prior methods either store the floating-point parameters (Yu et al., 2021; Du et al., 2020; Xu et al., 2022; Sun et al., 2024) to avoid accuracy degradation or abandon some integer values by replacing *rounding* with *floor*(Jin et al., 2020; Bulat & Tzimiropoulos, 2021) but leading to accuracy decline or training collapse at lower bit-widths. We propose *Double Rounding*, which applies the *rounding* operation twice instead of once, as shown in Figure1 (a). This approach ensures nearly lossless bit-switching and allows storing the highest bit-width model instead of the full-precision model. Specifically, the lower precision weight is included in the higher precision weight, reducing storage constraints.

Moreover, we empirically find severe competition between higher and lower precisions, particularly in 2-bit precision, as also noted in Tang et al. (2022); Xu et al. (2022). There are two reasons for this phenomenon: The optimal quantization interval itself is different for higher and lower precisions. Furthermore, shared weights are used for different precisions during joint training, but the quantization interval gradients for different precisions exhibit distinct magnitudes during training. Therefore, we introduce an Adaptive Learning Rate Scaling (ALRS) method, designed to dynamically adjust the learning rates across different precisions, which ensures consistent update steps of quantization scales corresponding to different precisions, as shown in the Figure 1 (b).

Finally, we develop an efficient one-shot mixed-precision quantization approach based on *Double Rounding*. Prior mixed-precision approaches first train a SuperNet with predefined bit-width lists, then search for optimal candidate SubNets under restrictive conditions, and finally retrain or fine-tune them, which incurs significant time and training costs. However, we use the Hessian Matrix Trace (Dong et al., 2020) as a sensitivity metric for different layers to optimize the SuperNet and propose a Hessian-Aware Stochastic Bit-switching (HASB) strategy, inspired by the Roulette algorithm (Dong et al., 2019a). This strategy enables tuned probability distribution of switching bit-width across layers, assigning higher bits to more sensitive layers and lower bits to less sensitive ones, as shown in Figure 1 (c). And, we add the sensitivity to the search stage as a constraint factor. So, our approach can omit the last stage. In conclusion, our main contributions can be described as:

- *Double Rounding* quantization method for multi-precision is proposed, which stores a single integer weight to enable adaptive precision switching with nearly lossless accuracy.

- Adaptive Learning Rate Scaling (ALRS) method for the multi-precision scheme is introduced, which effectively narrows the training convergence gap between high-precision and low-precision, enhancing the accuracy of low-precision models without compromising high-precision model accuracy.

- Hessian-Aware Stochastic Bit-switching (HASB) strategy for one-shot mixed-precision SuperNet is applied, where the access probability of bit-width for each layer is determined based on the layer's sensitivity.

- Experimental results on the ImageNet1K dataset demonstrate that our proposed methods are comparable to state-of-the-art methods across different mainstream CNN architectures.

## 2 RELATED WORKS

**Multi-Precision.** Multi-Precision entails a single shared model with multiple precisions by one-shot joint Quantization-Aware Training (QAT). This approach can dynamically adapt uniform bit-switching for the entire model according to computing resources and storage constraints. AdaBits (Jin et al., 2019) is the first work to consider adaptive bit-switching but encounters convergence issues with 2-bit quantization on ResNet50 (He et al., 2015). TQ (Zhang et al., 2021) quantizes weights or activation values by selecting a specific number of power-of-two terms. BitWave (Shi et al., 2024) is designed to leverage structured bit-level sparsity and dynamic dataflow to reduce computation and memory usage. Bit-Mixer (Bulat & Tzimiropoulos, 2021) addresses this problem by using the LSQ (Esser et al., 2019) quantization method but discards the lowest state quantized value, resulting in an accuracy decline. Multi-Precision joint QAT can also be viewed as a multi-objective optimization problem. Any-precision (Yu et al., 2021) and MultiQuant (Xu et al., 2022) combine knowledge distillation techniques to improve model accuracy. Among these methods, MultiQuant's proposed "Online Adaptive Label" training strategy is essentially a form of self-distillation (Kim et al., 2021). Similar to our method, AdaBits and Bit-Mixer can save an 8-bit model, while other methods rely on 32-bit models for bit switching. Our *Double Rounding* method can store the highest bit-width model (e.g., 8-bit) and achieve almost lossless bit-switching, ensuring a stable optimization process. Importantly, this leads to a reduction in training time by approximately 10% (Du et al., 2020) compared to separate quantization training.

**One-shot Mixed-Precision.** Previous works mainly utilize costly approaches, such as reinforcement learning (Wang et al., 2019; Elthakeb et al., 2019) and Neural Architecture Search (NAS) (Wu et al., 2018; Guo et al., 2020b; Shen et al., 2021), or rely on partial prior knowledge (Liu et al., 2021; Yao et al., 2021) for bit-width allocation, which may not achieve global optimality. In contrast, our proposed one-shot mixed-precision method employs Hessian-Aware optimization to refine a SuperNet via gradient updates, and then obtain the optimal conditional SubNets with less search cost without retraining or fine-tuning. Additionally, Bit-Mixer (Bulat & Tzimiropoulos, 2021) and MultiQuant (Xu et al., 2022) implement layer-adaptive mixed-precision models, but Bit-Mixer uses a naive search method to attain a sub-optimal solution, while MultiQuant requires 300 epochs of fine-tuning to achieve ideal performance. Unlike NAS approaches (Shen et al., 2021), which focus on altering network architecture (e.g., depth, kernel size, or channels), our method optimizes a once-for-all SuperNet using only quantization techniques without altering the model architecture.

## 3 METHODOLOGY

### 3.1 *Double Rounding*

Conventional separate precision quantization using Quantization-Aware Training (QAT) (Jacob et al., 2017) attain a fixed bit-width quantized model under a pre-trained FP32 model. A pseudo-quantization node is inserted into each layer of the model during training. This pseudo-quantization node comprises two operations: the quantization operation $quant(x)$, which maps floating-point (FP32) values to lower-bit integer values, and the dequantization operation $dequant(x)$, which restores the quantized integer value to its original floating-point representation. It can simulate the

quantization error incurred when compressing float values into integer values. As quantization involves a non-differentiable $Rounding$ operation, Straight-Through Estimator (STE) (Bengio et al., 2013) is commonly used to handle the non-differentiability.

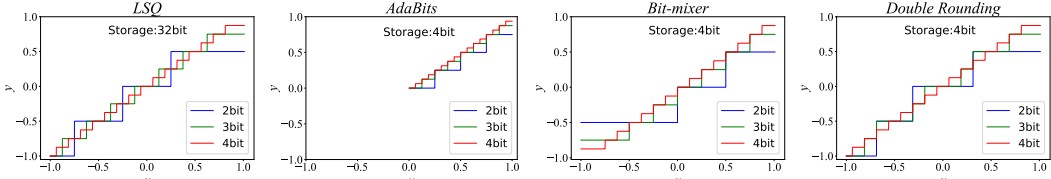

Figure 2: Comparison of four quantization schemes:(from left to right) used in *LSQ* (Esser et al., 2019), *AdaBits* (Jin et al., 2020), *Bit-Mixer* (Bulat & Tzimiropoulos, 2021) and Ours *Double Rounding*. In all cases $y = dequant(quant(x))$.

However, for multi-precision quantization, bit-switching can result in significant accuracy loss, especially when transitioning from higher bit-widths to lower ones, *e.g.*, from 8-bit to 2-bit. To mitigate this loss, prior works have mainly employed two strategies: one involves bit-switching from a floating-point model (32-bit) to a lower-bit model each time using multiple learnable quantization parameters, and the other substitutes the $Rounding$ operation with the $Floor$ operation, but this results in accuracy decline (especially in 2-bit). In contrast, we propose a nearly lossless bit-switching quantization method called *Double Rounding*. This method overcomes these limitations by employing a $Rounding$ operation twice. It allows the model to be saved in the highest-bit (*e.g.*, 8-bit) representation instead of full-precision, facilitating seamless switching to other bit-width models. A detailed comparison of *Double Rounding* with other quantization methods is shown in Figure 2.

Unlike AdaBits, which relies on the Dorefa (Zhou et al., 2016) quantization method where the quantization scale is determined based on the given bit-width, the quantization scale of our *Double Rounding* is learned online and is not fixed. It only requires a pair of shared quantization parameters, *i.e.*, *scale* and *zero-point*. Quantization scales of different precisions adhere to a strict "Power of Two" relationship. Suppose the highest-bit and the target low-bit are denoted as $h$-bit and $l$-bit respectively, and the difference between them is $\Delta = h - l$. The specific formulation of *Double Rounding* is as follows:

$$\widetilde{W_h} = \text{clip}(\left\lfloor \frac{W - \mathbf{z}_h}{\mathbf{s}_h} \right\rceil, -2^{h-1}, 2^{h-1} - 1) \tag{1}$$

$$\widetilde{W_l} = \text{clip}(\left\lfloor \frac{\widetilde{W_h}}{2^\Delta} \right\rceil, -2^{l-1}, 2^{l-1} - 1) \tag{2}$$

$$\widehat{W_l} = \widetilde{W_l} \times \mathbf{s}_h \times 2^\Delta + \mathbf{z}_h \tag{3}$$

where the symbol $\lfloor . \rceil$ denotes the $Rounding$ function, and $\text{clip}(x, low, upper)$ means $x$ is limited to the range between $low$ and $upper$. Here, $W$ represents the FP32 model's weights, $\mathbf{s}_h \in \mathbb{R}$ and $\mathbf{z}_h \in \mathbb{Z}$ denote the highest-bit (*e.g.*, 8-bit) quantization *scale* and *zero-point* respectively. $\widetilde{W_h}$ represent the quantized weights of the highest-bit, while $\widetilde{W_l}$ and $\widehat{W_l}$ represent the quantized weights and dequantized weights of the low-bit respectively.

Hardware shift operations can efficiently execute the division $\widetilde{W_h}$ by $1 << \Delta$. Note that in our *Double Rounding*, the model can also be saved at full precision by using unshared quantization parameters to run bit-switching and attain higher accuracy. Because we use symmetric quantization scheme, the $\mathbf{z}_h$ is 0. Please refer to Section A.4 for the gradient formulation of *Double Rounding*.

Unlike fixed weights, activations change online during inference. So, the corresponding *scale* and *zero-point* values for different precisions can be learned individually to increase overall accuracy. Suppose $X$ denotes the full precision activation, and $\widetilde{X_b}$ and $\widehat{X_b}$ are the quantized activation and dequantized activation respectively. The quantization process can be formulated as follows:

$$\widetilde{X_b} = \text{clip}(\left\lfloor \frac{X - \mathbf{z}_b}{\mathbf{s}_b} \right\rceil, 0, 2^b - 1) \tag{4}$$

$$\widehat{X_b} = \widetilde{X_b} \times \mathbf{s}_b + \mathbf{z}_b \tag{5}$$

where $\mathbf{s}_b \in \mathbb{R}$ and $\mathbf{z_b} \in \mathbb{Z}$ represent the quantization *scale* and *zero-point* of different bit-widths activation respectively. Note that $\mathbf{z}_b$ is 0 for the ReLU activation function.

## 3.2 Adaptive Learning Rate Scaling for Multi-Precision

Although our proposed *Double Rounding* method represents a significant improvement over most previous multi-precision works, the one-shot joint optimization of multiple precisions remains constrained by severe competition between the highest and lowest precisions (Tang et al., 2022; Xu et al., 2022). Different precisions simultaneously impact each other during joint training, resulting in substantial differences in convergence rates between them, as shown in Figure 3 (c). We experimentally find that this competitive relationship stems from the inconsistent magnitudes of the quantization scale's gradients between high-bit and low-bit quantization during joint training, as shown in Figure 3 (a) and (b). For other models statistical results please refer to Section A.6 in the appendix.

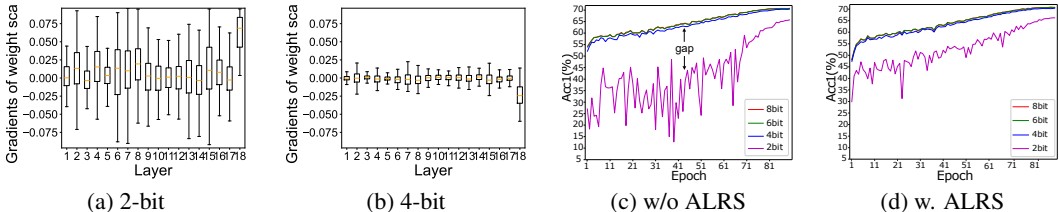

(a) 2-bit         (b) 4-bit         (c) w/o ALRS         (d) w. ALRS

Figure 3: The statistics of ResNet18 on ImageNet-1K dataset. (a) and (b): The quantization scale gradients' statistics for the weights, with outliers removed for clarity. (c) and (d): The multi-precision training processes of our *Double Rounding* without and with the ALRS strategy.

Motivated by these observations, we introduce a technique termed Adaptive Learning Rate Scaling (ALRS), which dynamically adjusts learning rates for different precisions to optimize the training process. This technique is inspired by the Layer-wise Adaptive Rate Scaling (LARS) (You et al., 2017) optimizer. Specifically, suppose the current batch iteration's learning rate is $\lambda$, we set learning rates $\lambda_b$ of different precisions as follows:

$$\lambda_b = \eta_b \left( \lambda - \sum_{i=1}^{L} \frac{\min \left( \text{max\_abs} \left( \text{clip\_grad}(\nabla \mathbf{s}_b^i, 1.0) \right), 1.0 \right)}{L} \right), \tag{6}$$

$$\eta_b = \begin{cases} 1 \times 10^{-\frac{\Delta}{2}}, & \text{if } \Delta \text{ is even} \\ 5 \times 10^{-(\frac{\Delta+1}{2})}, & \text{if } \Delta \text{ is odd} \end{cases} \tag{7}$$

where the $L$ is the number of layers, clip_grad(.) represents gradient clipping that prevents gradient explosion, max_abs(.) denotes the maximum absolute value of all elements. The $\nabla \mathbf{s}_b^i$ denotes the quantization scale's gradients of layer $i$ and $\eta_b$ denotes scaling hyperparameter of different precisions, *e.g.*, 8-bit is 1, 6-bit is 0.1, and 4-bit is 0.01. Note that the ALRS strategy is only used for updating quantization scales. It can adaptively update the learning rates of different precisions and ensure that model can optimize quantization parameters at the same pace, ultimately achieving a minimal convergence gap in higher bits and 2-bit, as shown in Figure 3 (d).

In multi-precision scheme, different precisions share the same model weights during joint training. For conventional multi-precision, the shared weight computes $n$ forward processes at each training iteration, where $n$ is the number of candidate bit-widths. The losses attained from different precisions are then accumulated, and the gradients are computed. Finally, the shared parameters are updated. For detailed implementation please refer to Algorithm A.1 in the appendix. However, we find that if different precision losses separately compute gradients and directly update shared parameters at each forward process, it attains better accuracy when combined with our ALRS training strategy. Additionally, we use dual optimizers to update the weight parameters and quantization parameters simultaneously. We also set the weight-decay of the quantization scales to 0 to achieve stable convergence. For detailed implementation please refer to Algorithm A.2 in the appendix.

## 3.3 One-Shot Mixed-Precision SuperNet

Unlike multi-precision, where all layers uniformly utilize the same bit-width, mixed-precision SuperNet provides finer-grained adaptive by configuring the bit-width at different layers. Previous methods typically decouple the training and search stages, which need a third stage for retraining

or fine-tuning the searched SubNets. These approaches generally incur substantial search costs in selecting the optimal SubNets, often employing methods such as greedy algorithms (Cai & Vasconcelos, 2020; Bulat & Tzimiropoulos, 2021) or genetic algorithms (Guo et al., 2020a; Xu et al., 2022). Considering the fact that the sensitivity (Dong et al., 2019b), *i.e.*, importance, of each layer is different, we propose a Hessian-Aware Stochastic Bit-switching (HASB) strategy for one-shot mixed-precision training.

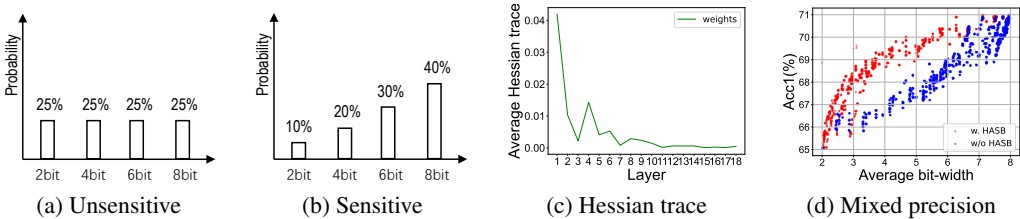

| (a) Unsensitive | (b) Sensitive | (c) Hessian trace | (d) Mixed precision |

Figure 4: The HASB stochastic process and Mixed-precision of ResNet18 for {2,4,6,8}-bit.

Specifically, the Hessian Matrix Trace (HMT) is utilized to measure the sensitivity of each layer. We first need to compute the pre-trained model's HMT by around 1000 training images (Dong et al., 2020), as shown in Figure 4 (c). Then, the HMT of different layers is utilized as the probability metric for bit-switching. Higher bits are priority selected for sensitive layers, while all candidate bits are equally selected for unsensitive layers. Our proposed Roulette algorithm is used for bit-switching processes of different layers during training, as shown in the Algorithm 1. If a layer's HMT exceeds the average HMT of all layers, it is recognized as sensitive, and the probability distribution of Figure 4 (b) is used for bit selection. Conversely, if the HMT is below the average, the probability distribution of Figure 4 (a) is used for selection. Finally, the Integer Linear Programming (ILP) (Ma et al., 2023) algorithm is employed to find the optimal SubNets. Considering each layer's sensitivity during training and adding this sensitivity to the ILP's constraint factors (*e.g.*, model's FLOPs, latency, and parameters), which depend on the actual deployment requirements. We can efficiently attain a set of optimal SubNets during the search stage without retraining, thereby significant reduce the overall costs. All the searched SubNets collectively constitute the Pareto Frontier optimal solution, as shown in Figure 4 (d). For detailed mixed-precision training and searching process (*i.e.*, ILP) please refer to the Algorithm A.3 and the Algorithm 2 respectively. Note that the terms "pulp" and "index" in Algorithm 2 represent a Python library for linear programming optimization and the position of the maximum bit-width in the candidate bit-widths, respectively.

---

**Algorithm 1** Roulette algorithm in bit-switching

**Require:** Candidate bit-widths set $b \in B$, the HMT of current layer: $t_l$, average HMT: $t_m$;
1: Sample $r \sim U(0, 1]$ from a uniform distribution;
2: **if** $t_l < t_m$ **then**
3:     Compute bit-switching probability of all candidate $b_i$ with $p_i = 1/n$;
4:     Set $s = 0$, and $i = 0$;
5:     **while** $s < r$ **do**
6:         $i = i + 1$;
7:         $s = p_i + s$;
8:     **end while**
9: **else**
10:    Compute bit-switching probability of all candidate $b_i$ with $p_i = b_i/\|B\|_1$;
11:    Set $s = 0$, and $i = 0$;
12:    **while** $s < r$ **do**
13:       $i = i + 1$;
14:       $s = p_i + s$;
15:    **end while**
16: **end if**
17: **return** $b_i$;

Note that $n$ and $L$ represent the number of candidate bit-widths and model layers respectively, and $\| \cdot \|_1$ is $L_1$ norm.

---

**Algorithm 2** Our searching process for SubNets

**Input:** Candidate bit-widths set $b \in B$, the HMT of different layers of FP32 model: $t_l \in \{T\}_{l=1}^L$, the constraint average bit-width: $\omega$, each layer parameters: $n_l \in \{N\}_{l=1}^L$;
1: Initial searched SubNets'solutions: $S = \phi$
2: Minimal objective : $O = \sum_{l=1}^L \frac{t_l}{n_l} \cdot b_l$
3: Constraints: $\omega \equiv \frac{\sum_{l=1}^L b_l}{L}$
4: The first solve: $\mathbf{s_1} = pulp.solve(O, \omega)$ and $S.append(\mathbf{s_1})$
5: **for** $c_i$ in $\mathbf{s_1}$ **do**
6:    **for** $b$ in $B[: index(max(\mathbf{s_1}))]$ **do**
7:       **if** $b \neq c_i$ **then**
8:          Add constraint: $b \equiv c_i$
9:          Solve: $\mathbf{s} = pulp.solve(O, \omega, b)$
10:         **if** $\mathbf{s}$ not in $S$ **then**
11:            $S.append(\mathbf{s})$
12:         **end if**
13:         Pop last constraint: $b \equiv c_i$
14:       **end if**
15:    **end for**
16: **end for**
17: **return** $S$

---

# 4 EXPERIMENTAL RESULTS

## 4.1 IMAGE CLASSIFICATION

**Setup.** In this paper, we mainly focus on ImageNet-1K classification task using both classical networks (ResNet18/50) and lightweight networks (MobileNetV2), which same as previous works. Experiments cover joint quantization training for multi-precision and mixed precision. We explore two candidate bit configurations, *i.e.*, {8,6,4,2}-bit and {4,3,2}-bit, each number represents the quantization level of the weight and activation layers. Like previous methods, we exclude batch normalization layers from quantization, and the first and last layers are kept at full precision. We initialize the multi-precision models with a pre-trained FP32 model, and initialize the mixed-precision models with a pre-trained multi-precision model. All models use the *Adam* optimizer with a batch size of 256 for 90 epochs and use a cosine scheduler without warm-up phase. The initial learning rate is 5e-4 and weight decay is 5e-5. Data augmentation uses the standard set of transformations including random cropping, resizing to 224×224 pixels, and random flipping. Images are resized to 256×256 pixels and then center-cropped to 224×224 resolution during evaluation.

### 4.1.1 MULTI-PRECISION

**Results.** For {8,6,4,2}-bit configuration, the Top-1 validation accuracy is shown in Table 1. The network weights and the corresponding activations are quantized into w-bit and a-bit respectively. Our *double-rounding* combined with ALRS training strategy surpasses the previous state-of-the-art (SOTA) methods. For example, in ResNet18, it exceeds Any-Precision by 2.7%(or 2.83%) under w8a8 setting without(or with) using KD technique, and outperforms MultiQuant by 0.63%(or 0.73%) under w4a4 setting without(or with) using KD technique respectively. Additionally, when the candidate bit-list includes 2-bit, the previous methods can't converge on MobileNetV2 during training. So, they use {8,6,4}-bit precision for MobileNetV2 experiments. For consistency, we also test {8,6,4}-bit results, as shown in the "Ours {8,6,4}-bit" rows of Table 1. Our method achieves 0.25%/0.11%/0.56% higher accuracy than AdaBits under the w8a8/w6a6/w4a4 settings.

Notably, our method exhibits the ability to converge but shows a big decline in accuracy on MobileNetV2. On the one hand, the compact model exhibits significant differences in the quantization scale gradients of different channels due to involving Depth-Wise Convolution (Sheng et al., 2018). On the other hand, when the bit-list includes 2-bit, it intensifies competition between different precisions during training. To improve the accuracy of compact models, we suggest considering the per-layer or per-channel learning rate scaling techniques in future work.

Table 1: Top1 accuracy comparisons on multi-precision of {8,6,4,2}-bit on ImageNet-1K datasets. "KD" denotes knowledge distillation. The "−" represents the unqueried value.

| Model | Method | KD | Storage | Epoch | w8a8 | w6a6 | w4a4 | w2a2 | FP |
|---|---|---|---|---|---|---|---|---|---|
| ResNet18 | Hot-Swap(Sun et al., 2021) | ✗ | 32bit | − | 70.40 | 70.30 | 70.20 | 64.90 | − |
| | L1(Alizadeh et al., 2020) | ✗ | 32bit | − | 69.92 | 66.39 | 0.22 | − | 70.07 |
| | KURE(Chmiel et al., 2020) | ✗ | 32bit | 80 | 70.20 | 70.00 | 66.90 | − | 70.30 |
| | Ours | ✗ | 8bit | 90 | 70.74 | 70.71 | 70.43 | 66.35 | 69.76 |
| | Any-Precision(Yu et al., 2021) | ✓ | 32bit | 80 | 68.04 | − | 67.96 | 64.19 | 69.27 |
| | CoQuant(Du et al., 2020) | ✓ | 8bit | 100 | 67.90 | 67.60 | 66.60 | 57.10 | 69.90 |
| | MultiQuant(Xu et al., 2022) | ✓ | 32bit | 90 | 70.28 | 70.14 | 69.80 | 66.56 | 69.76 |
| | Ours | ✓ | 8bit | 90 | **70.87** | **70.79** | **70.53** | **66.84** | 69.76 |
| ResNet50 | Any-Precision(Yu et al., 2021) | ✗ | 32bit | 80 | 74.68 | − | 74.43 | 72.88 | 75.95 |
| | Hot-Swap(Sun et al., 2021) | ✗ | 32bit | − | 75.60 | 75.50 | 75.30 | 71.90 | − |
| | KURE(Chmiel et al., 2020) | ✗ | 32bit | 80 | − | 76.20 | 74.30 | − | 76.30 |
| | Ours | ✗ | 8bit | 90 | 76.51 | 76.28 | 75.74 | 72.31 | 76.13 |
| | Any-Precision(Yu et al., 2021) | ✓ | 32bit | 80 | 74.91 | − | 74.75 | 73.24 | 75.95 |
| | MultiQuant(Xu et al., 2022) | ✓ | 32bit | 90 | 76.94 | 76.85 | 76.46 | 73.76 | 76.13 |
| | Ours | ✓ | 8bit | 90 | **76.98** | **76.86** | **76.52** | **73.78** | 76.13 |
| MobileNetV2 | AdaBits(Jin et al., 2020) | ✗ | 8bit | 150 | 72.30 | 72.30 | 70.30 | − | 71.80 |
| | KURE(Chmiel et al., 2020) | ✗ | 32bit | 80 | − | 70.00 | 59.00 | − | 71.30 |
| | Ours {8,6,4}-bit | ✗ | 8bit | 90 | 72.42 | 72.06 | 69.92 | − | 71.14 |
| | MultiQuant(Xu et al., 2022) | ✓ | 32bit | 90 | 72.33 | 72.09 | 70.59 | − | 71.88 |
| | Ours {8,6,4}-bit | ✓ | 8bit | 90 | **72.55** | **72.41** | **70.86** | − | 71.14 |
| | Ours {8,6,4,2}-bit | ✗ | 8bit | 90 | 70.98 | 70.70 | 68.77 | 50.43 | 71.14 |
| | Ours {8,6,4,2}-bit | ✓ | 8bit | 90 | 71.35 | 71.20 | 69.85 | **53.06** | 71.14 |

For {4,3,2}-bit configuration, Table 2 demonstrate that our *double-rounding* consistently surpasses previous SOTA methods. For instance, in ResNet18, it exceeds Bit-Mixer by 0.63%/0.7%/1.2%(or 0.37%/0.64%/1.02%) under w4a4/w3a3/w2a2 settings without(or with) using KD technique, and outperforms ABN by 0.87%/0.74%/1.12% under w4a4/w3a3/w2a2 settings with using KD technique respectively. In ResNet50, Our method outperforms Bit-Mixer by 0.86%/0.63%/0.1% under w4a4/w3a3/w2a2 settings.

Notably, the overall results of Table 2 are worse than the {8,6,4,2}-bit configuration for joint training. We analyze that this discrepancy arises from information loss in the shared lower precision model (*i.e.*, 4-bit) used for bit-switching. In other words, compared with 4-bit, it is easier to directly optimize 8-bit quantization parameters to converge to the optimal value. So, we recommend including 8-bit for multi-precision training. Furthermore, independently learning the quantization scales for different precisions, including weights and activations, significantly improves accuracy compared to using shared scales. However, it requires saving the model in 32-bit format, as shown in "Ours*" of Table 2.

Table 2: Top1 accuracy comparisons on multi-precision of {4,3,2}-bit on ImageNet-1K datasets.

| Model | Method | KD | Storage | Epoch | w4a4 | w3a3 | w2a2 | FP |
|---|---|---|---|---|---|---|---|---|
| ResNet18 | Bit-Mixer(Bulat & Tzimiropoulos, 2021) | ✗ | 4bit | 160 | 69.10 | 68.50 | 65.10 | 69.60 |
| | Vertical-layer(Wu et al., 2023) | ✗ | 4bit | 300 | 69.20 | 68.80 | 66.60 | 70.50 |
| | Ours | ✗ | 4bit | 90 | 69.73 | 69.20 | 66.30 | 69.76 |
| | Q-DNNs(Du et al., 2020) | ✓ | 32bit | 45 | 66.94 | 66.28 | 62.91 | 68.60 |
| | ABN(Tang et al., 2022) | ✓ | 4bit | 160 | 68.90 | 68.60 | 65.50 | — |
| | Bit-Mixer(Bulat & Tzimiropoulos, 2021) | ✓ | 4bit | 160 | 69.40 | 68.70 | 65.60 | 69.60 |
| | Ours | ✓ | 4bit | 90 | **69.77** | **69.34** | **66.62** | 69.76 |
| ResNet50 | Ours | ✗ | 4bit | 90 | 75.81 | 75.24 | 71.62 | 76.13 |
| | AdaBits(Jin et al., 2020) | ✗ | 32bit | 150 | 76.10 | 75.80 | 73.20 | 75.00 |
| | Ours* | ✗ | 32bit | 90 | **76.42** | **75.82** | **73.28** | 76.13 |
| | Bit-Mixer(Bulat & Tzimiropoulos, 2021) | ✓ | 4bit | 160 | 75.20 | 74.90 | 72.70 | — |
| | Ours | ✓ | 4bit | 90 | 76.06 | 75.53 | 72.80 | 76.13 |

### 4.1.2 MIXED-PRECISION

**Results.** We follow previous works to conduct mixed-precision experiments based on the {4,3,2}-bit configuration. Our proposed one-shot mixed-precision joint quantization method with the HASB technique comparable to the previous SOTA methods, as presented in Table 3. For example, in ResNet18, our method exceeds Bit-Mixer by 0.83%/0.72%/0.77%/7.07% under w4a4/w3a3/w2a2/3MP settings and outperforms EQ-Net (Xu et al., 2023) by 0.2% under 3MP setting. The results demonstrate the effectiveness of one-shot mixed-precision joint training to consider sensitivity with Hessian Matrix Trace when randomly allocating bit-widths for different layers. Additionally, Table 3 reveals that our results do not achieve optimal performance across all settings. We hypothesize that extending the number of training epochs or combining ILP with other efficient search methods, such as genetic algorithms, may be necessary to achieve optimal results in mixed-precision optimization.

Table 3: Top1 accuracy comparisons on mixed-precision of {4,3,2}-bit on ImageNet-1K dataset. "MP" denotes average bit-width for mixed-precision. The "−" represents the unqueried value.

| Model | Method | KD | Training | Searching | Fine-tune | Epoch | w4a4 | w3a3 | w2a2 | 3MP | FP |
|---|---|---|---|---|---|---|---|---|---|---|---|
| ResNet18 | Ours | ✗ | HASB | ILP | w/o | 90 | 69.80 | 68.63 | 64.88 | 68.85 | 69.76 |
| | Bit-Mixer | ✓ | Random | Greedy | w/o | 160 | 69.20 | 68.60 | 64.40 | 62.90 | 69.60 |
| | ABN | ✓ | DRL | DRL | w. | 160 | 69.80 | 69.00 | **66.20** | 67.70 | — |
| | MultiQuant | ✓ | LRH | Genetic | w. | 90 | — | 67.50 | — | 69.20 | 69.76 |
| | EQ-Net | ✓ | LRH | Genetic | w. | 120 | — | 69.30 | 65.90 | 69.80 | 69.76 |
| | Ours | ✓ | HASB | ILP | w/o | 90 | **70.03** | **69.32** | 65.17 | **69.92** | 69.76 |
| ResNet50 | Ours | ✗ | HASB | ILP | w/o | 90 | 75.01 | 74.31 | 71.47 | 75.06 | 76.13 |
| | Bit-Mixer | ✓ | Random | Greedy | w/o | 160 | 75.20 | **74.80** | 72.10 | 73.20 | — |
| | EQ-Net | ✓ | LRH | Genetic | w. | 120 | — | 74.70 | **72.50** | 75.10 | 76.13 |
| | Ours | ✓ | HASB | ILP | w/o | 90 | **75.63** | 74.36 | 72.32 | **75.24** | 76.13 |

## 4.2 OBJECT DETECTION AND SEGMENTATION

**Setup.** We utilize the pre-trained models as the backbone within the Mask-RCNN (He et al., 2017) detector for object detection and instance segmentation on the MS-COCO 2017 dataset, comprising

118K training images and 5K validation images. We follow the Pytorch official "code" and employ the AdamW optimizer, conduct training of 26 epochs, use a batch size of 16, and maintain other training settings without further hyperparameter tuning.

**Results.** Table 4 reports the average precision (mAP) performance for both detection and instance segmentation tasks for quantizing the backbone of the Mask-RCNN model on the COCO dataset. The results further confirm the generalization capabilities of our *Double Rounding*.

Table 4: Results of multi-precision on object detection and instance segmentation benchmark.

| Backbone | $mAP^b_{FP}$ | $mAP^b_{w8a8}$ | $mAP^b_{w6a6}$ | $mAP^b_{w4a4}$ | $mAP^b_{w2a2}$ | $mAP^m_{FP}$ | $mAP^m_{w8a8}$ | $mAP^m_{w6a6}$ | $mAP^m_{w4a4}$ | $mAP^m_{w2a2}$ |
|---|---|---|---|---|---|---|---|---|---|---|
| ResNet18 | 27.3 | 27.8 | 27.1 | 26.5 | 21.3 | 25.6 | 25.8 | 25.0 | 24.6 | 20.7 |
| ResNet50 | 37.9 | 37.0 | 36.6 | 34.8 | 26.4 | 34.6 | 33.7 | 32.4 | 31.7 | 25.0 |

## 4.3 LLMS TASK

We also conduct experiments on Large Language Models (LLMs) to validate the effectiveness of our method in more recent architectures, as shown in Table 5. We conduct multi-precision experiments on small LLMs (Zhang et al., 2024) without using ALRS and distillation. Note that, except for not quantizing the embedding layer and head layer, due to the sensitivity of the SiLU activation causing non-convergence, we don't quantize the SiLU activation in the MLP and set the batch size to 16. The results demonstrate that our approach applies to more recent and complex models.

Table 5: Zero-shot performance on commonsense reasoning tasks under different LLM models.

| Model | Precision | HellaSwag | Obqa | WinoGrande | ARC-c | ARC-e | boolq | piqa | Avg. ↑ |
|---|---|---|---|---|---|---|---|---|---|
| TinyLlama 120M | FP | 26.07 | 27.20 | 49.64 | 28.58 | 25.51 | 47.86 | 49.73 | 36.37 |
| | w8a8 | 26.33 | 28.00 | 48.15 | 28.84 | 26.14 | 62.17 | 50.05 | **38.53** |
| iteration-step 4000 | w6a6 | 26.29 | 26.60 | 49.80 | 27.65 | 27.69 | 56.45 | 50.65 | 37.88 |
| | w4a4 | 26.12 | 26.00 | 48.62 | 29.27 | 26.52 | 47.55 | 49.84 | 36.27 |
| | w2a2 | 26.17 | 25.20 | 49.72 | 29.01 | 26.09 | 50.43 | 49.13 | 36.54 |
| | w8a8 | 26.31 | 28.20 | 48.62 | 28.75 | 26.18 | 62.17 | 50.05 | **38.61** |
| iteration-step 8000 | w6a6 | 26.52 | 26.40 | 49.96 | 29.01 | 27.61 | 49.63 | 49.89 | 37.00 |
| | w4a4 | 25.89 | 26.60 | 49.80 | 28.67 | 26.43 | 43.03 | 50.33 | 35.82 |
| | w2a2 | 25.83 | 24.00 | 50.83 | 28.41 | 26.05 | 44.01 | 50.60 | 35.68 |
| TinyLlama 1.1B | FP | 59.20 | 36.00 | 59.12 | 30.12 | 55.25 | 57.83 | 73.29 | 52.99 |
| | w8a8 | 57.58 | 35.60 | 58.48 | 28.26 | 51.39 | 62.87 | 72.31 | **52.36** |
| iteration-step 4000 | w6a6 | 52.36 | 31.00 | 57.62 | 26.71 | 47.35 | 59.39 | 69.15 | 49.08 |
| | w4a4 | 25.71 | 24.60 | 49.64 | 27.82 | 25.76 | 49.88 | 49.13 | 36.08 |
| | w2a2 | 25.73 | 27.60 | 51.22 | 26.54 | 25.72 | 62.17 | 50.27 | 38.46 |
| | w8a8 | 57.79 | 35.60 | 58.72 | 30.20 | 53.24 | 62.69 | 72.14 | **52.91** |
| iteration-step 8000 | w6a6 | 51.57 | 30.00 | 57.77 | 25.34 | 46.76 | 56.85 | 68.39 | 48.10 |
| | w4a4 | 25.93 | 24.60 | 51.85 | 28.16 | 25.29 | 51.59 | 49.89 | 36.76 |
| | w2a2 | 25.81 | 27.40 | 51.22 | 26.45 | 25.93 | 62.17 | 50.16 | 38.45 |

## 4.4 ABLATION STUDIES

**ALRS vs. Conventional in Multi-Precision.** To verify the effectiveness of our proposed ALRS training strategy, we conduct an ablation experiment without KD, as shown in Table 6, and observe overall accuracy improvements, particularly for the 2bit. Like previous works, where MobileNetV2 can't achieve stable convergence with {4,3,2}-bit, we also opt for {8,6,4}-bit to keep consistent. However, our method can achieve stable convergence with {8,6,4,2}-bit quantization. This demonstrates the superiority of our proposed *Double-Rounding* and ALRS methods. In addition, we conduct ablation studies of other methods with or without ALRS, as shown in Table 7. The results further validate the effectiveness of our proposed ALRS for multi-precision.

**Multi-Precision vs. Separate-Precision in Time Cost.** We statistic the results regarding the time cost for normal multi-precision compared to separate-precision quantization, as shown in Table 8. Multi-precision training costs stay approximate constant as the number of candidate bit-widths.

**Pareto Frontier of Different Mixed-Precision Configurations.** To verify the effectiveness of our HASB strategy, we conduct ablation experiments on different bit-lists. Figure 5 shows the search results of Mixed-precision SuperNet under {8,6,4,2}-bit, {4,3,2}-bit and {8,4}-bit configurations respectively. Where each point represents a SubNet. These results are obtained directly from ILP

Table 6: Ablation studies of multi-precision, ResNet20 on CIFAR-10 dataset and other models on ImageNet-1K dataset. Note that MobileNetV2 uses {8,6,4}-bit instead of {4,3,2}-bit.

| Model | ALRS | w8a8 | w6a6 | w4a4 | w2a2 | w4a4 | w3a3 | w2a2 | FP |
|---|---|---|---|---|---|---|---|---|---|
| ResNet20 | w/o | 92.17 | 92.20 | 92.17 | 89.67 | 91.19 | 90.98 | 88.62 | 92.30 |
|  | w. | 92.25 | 92.32 | 92.09 | 90.19 | 91.79 | 91.83 | 88.88 | 92.30 |
| ResNet18 | w/o | 70.05 | 69.80 | 69.32 | 65.83 | 69.38 | 68.74 | 65.62 | 69.76 |
|  | w. | 70.74 | 70.71 | 70.43 | 66.35 | 69.73 | 69.20 | 66.30 | 69.76 |
| ResNet50 | w/o | 76.18 | 76.08 | 75.64 | 70.28 | 75.48 | 74.85 | 70.64 | 76.13 |
|  | w. | 76.51 | 76.28 | 75.74 | 72.31 | 75.81 | 75.24 | 71.62 | 76.13 |
|  |  | w8a8 | w6a6 | w4a4 | w2a2 | w8a8 | w6a6 | w4a4 |  |
| MobileNetV2 | w/o | 70.55 | 70.65 | 68.08 | 45.00 | 72.06 | 71.87 | 69.40 | 71.14 |
|  | w. | 70.98 | 70.70 | 68.77 | 50.43 | 72.42 | 72.06 | 69.92 | 71.14 |

Table 7: Ablation studies of Multi-Precision with or without ALRS strategy for other methods.

| Model | Dataset | Method | ALRS | Storage | Epoch | w8a8 | w6a6 | w4a4 | w2a2 | FP |
|---|---|---|---|---|---|---|---|---|---|---|
| ResNet20 | Cifar10 | Bit-Mixer | w/o | 8bit | 90 | 91.84 | 91.89 | 91.34 | 38.19 | 92.30 |
|  |  | Bit-Mixer | w. | 8bit | 90 | **92.07** | 91.88 | **91.97** | 59.08 | 92.30 |
|  |  | MultiQuant | w/o | 32bit | 90 | 92.02 | 91.89 | 91.50 | 87.78 | 92.30 |
|  |  | MultiQuant | w. | 32bit | 90 | 92.04 | **92.08** | 91.56 | **88.50** | 92.30 |
| ResNet18 | ImageNet | Bit-Mixer | w/o | 8bit | 90 | 70.24 | 70.16 | 68.60 | 62.64 | 69.76 |
|  |  | Bit-Mixer | w. | 8bit | 90 | 70.36 | 70.28 | 69.43 | 63.12 | 69.76 |
|  |  | MultiQuant | w/o | 32bit | 90 | 70.56 | 70.64 | 70.21 | 66.05 | 69.76 |
|  |  | MultiQuant | w. | 32bit | 90 | **70.85** | **70.75** | **70.46** | **66.43** | 69.76 |

Table 8: Training costs for multi-precision and separate-precision are averaged over three runs.

| Model | Dataset | Bit-widths | #V100 | Epochs | BatchSize | Avg. hours | Save cost (%) |
|---|---|---|---|---|---|---|---|
| ResNet20 | Cifar10 | Separate-bit | 1 | 200 | 128 | 0.9 | 0.0 |
|  |  | {4,3,2}-bit | 1 | 200 | 128 | 0.7 | 28.6 |
|  |  | {8,6,4,2}-bit | 1 | 200 | 128 | 0.8 | 12.5 |
| ResNet18 | ImageNet | Separate-bit | 4 | 90 | 256 | 19.0 | 0.0 |
|  |  | {4,3,2}-bit | 4 | 90 | 256 | 15.2 | 25.0 |
|  |  | {8,6,4,2}-bit | 4 | 90 | 256 | 16.3 | 16.6 |
| ResNet50 | ImageNet | Separate-bit | 4 | 90 | 256 | 51.6 | 0.0 |
|  |  | {4,3,2}-bit | 4 | 90 | 256 | 40.7 | 26.8 |
|  |  | {8,6,4,2}-bit | 4 | 90 | 256 | 40.8 | 26.5 |

sampling without retraining or fine-tuning. As the figure shows, the highest red points are higher than the blue points under the same bit width, indicating that this strategy is effective.

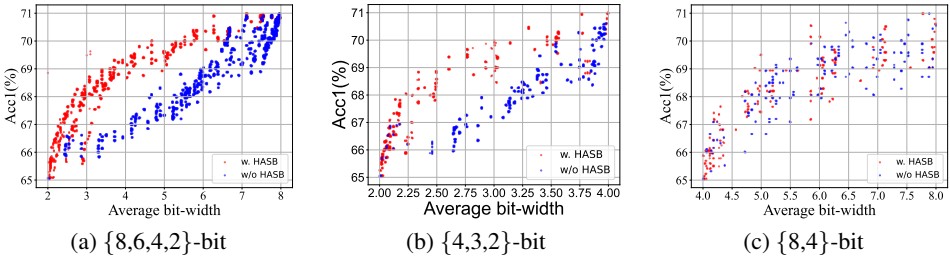

(a) {8,6,4,2}-bit      (b) {4,3,2}-bit      (c) {8,4}-bit

Figure 5: Comparison of HASB and Baseline approaches for Mixed-Precision on ResNet18.

## 5 CONCLUSION

This paper first introduces *Double Rounding* quantization method used to address the challenges of multi-precision and mixed-precision joint training. It can store single integer-weight parameters and attain nearly lossless bit-switching. Secondly, we propose an Adaptive Learning Rate Scaling (ALRS) method for multi-precision joint training that narrows the training convergence gap between high-precision and low-precision, enhancing model accuracy of multi-precision. Finally, our proposed Hessian-Aware Stochastic Bit-switching (HASB) strategy for one-shot mixed-precision SuperNet and efficient searching method combined with Integer Linear Programming, achieving approximate Pareto Frontier optimal solution. Our proposed methods aim to achieve a flexible and effective model compression technique for adapting different storage and computation requirements.

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

# A APPENDIX

## A.1 OVERVIEW

In this supplementary material, we present more explanations and experimental results.

- First, we provide a detailed explanation of the different quantization types under QAT.

- We then present a comparison of multi-precision and separate-precision on the ImageNet-1k dataset.

- Furthermore, we provide the gradient formulation of Double Rounding.

- And, the algorithm implementation of both multi-precision and mixed-precision training approaches.

- Then, we provide more gradient statistics of learnable quantization scales in different networks.

- Finally, we also provide the bit-widths learned by each layer of the mixed-precision with a given average bit-width condition.

## A.2 DIFFERENT QUANTIZATION TYPES

In this section, we provide a detailed explanation of the different quantization types during Quantization-Aware Training (QAT), as is shown in Figure 6.

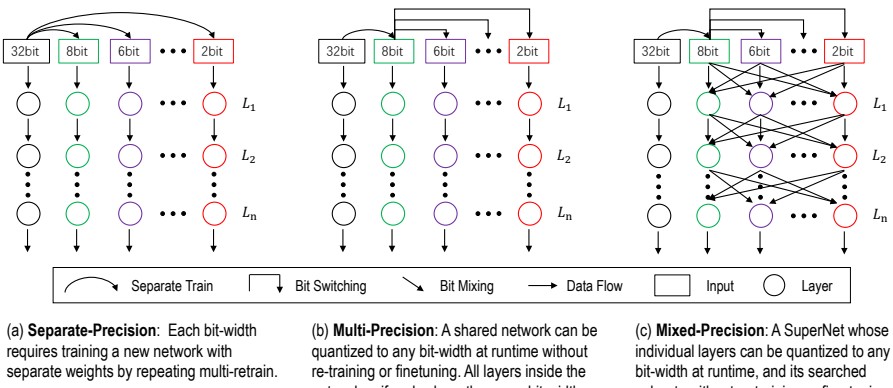

(a) **Separate-Precision**: Each bit-width requires training a new network with separate weights by repeating multi-retrain.

(b) **Multi-Precision**: A shared network can be quantized to any bit-width at runtime without re-training or finetuning. All layers inside the network uniformly share the same bit-width.

(c) **Mixed-Precision**: A SuperNet whose individual layers can be quantized to any bit-width at runtime, and its searched subnets without re-training or fine-tuning.

Figure 6: Comparison between different quantization types during quantization-aware training.

## A.3 MULTI-PRECISION VS. SEPARATE-PRECISION.

We provide the comparison of Multi-Precision and Separate-Precision on ImageNet-1K dataset. Table 9 shows that our Multi-Precision joint training scheme has comparable accuracy of different precisions compared to Separate-Precision with multiple re-train. This further proves the effectiveness of our proposed One-shot *Double Rounding* Multi-Precision method.

Table 9: Top1 accuracy comparisons on multi-precision of {8,6,4,2}-bit on ImageNet-1K datasets.

| Model | Method | One-shot | Storage | Epoch | w8a8 | w6a6 | w4a4 | w2a2 | FP |
|---|---|---|---|---|---|---|---|---|---|
| ResNet18 | LSQ(Esser et al., 2019) | ✗ | {8,6,4,2}-bit | 90 | **71.10** | − | **71.10** | **67.60** | 70.50 |
| | LSQ+(Bhalgat et al., 2020) | ✗ | {8,6,4,2}-bit | 90 | − | − | 70.80 | 66.80 | 70.10 |
| | Ours | ✓ | 8-bit | 90 | 70.74 | 70.71 | 70.43 | 66.35 | 69.76 |
| ResNet50 | LSQ(Esser et al., 2019) | ✗ | {8,6,4,2}-bit | 90 | **76.80** | − | **76.70** | **73.70** | 76.90 |
| | Ours | ✓ | 8-bit | 90 | 76.51 | 76.28 | 75.74 | 72.31 | 76.13 |

### A.4 THE GRADIENT FORMULATION OF DOUBLE ROUNDING

A general formulation for uniform quantization process is as follows:

$$\widetilde{W} = \text{clip}\left(\left\lfloor \frac{W}{\mathbf{s}} \right\rceil + \mathbf{z}, -2^{b-1}, 2^{b-1} - 1\right) \tag{8}$$

$$\widehat{W} = (\widetilde{W} - \mathbf{z}) \times \mathbf{s} \tag{9}$$

where the symbol $\lfloor . \rceil$ denotes the *Rounding* function, $\text{clip}(x, low, upper)$ expresses $x$ below $low$ are set to $low$ and above $upper$ are set to $upper$. $b$ denotes the quantization level (or bit-width), $\mathbf{s} \in \mathbb{R}$ and $\mathbf{z} \in \mathbb{Z}$ represents the quantization *scale* (or interval) and *zero-point* associated with each $b$, respectively. $W$ represents the FP32 model's weights, $\widetilde{W}$ signifies the quantized integer weights, and $\widehat{W}$ represents the dequantized floating-point weights.

The quantization scale of our *Double Rounding* is learned online and not fixed. And it only needs a pair of shared quantization parameters, *i.e.*, *scale* and *zero-point*. Suppose the highest-bit and the low-bit are denoted as $h$-bit and $l$-bit respectively, and the difference between them is $\Delta = h - l$. The specific formulation is as follows:

$$\widetilde{W}_h = \text{clip}\left(\left\lfloor \frac{W - \mathbf{z}_h}{\mathbf{s}_h} \right\rceil, -2^{h-1}, 2^{h-1} - 1\right) \tag{10}$$

$$\widetilde{W}_l = \text{clip}\left(\left\lfloor \frac{\widetilde{W}_h}{2^\Delta} \right\rceil, -2^{l-1}, 2^{l-1} - 1\right) \tag{11}$$

$$\widehat{W}_l = \widetilde{W}_l \times \mathbf{s}_h \times 2^\Delta + \mathbf{z}_h \tag{12}$$

where $\mathbf{s}_h \in \mathbb{R}$ and $\mathbf{z}_h \in \mathbb{Z}$ denote the highest-bit quantization *scale* and *zero-point* respectively. $\widetilde{W}_h$ and $\widetilde{W}_l$ represent the quantized weights of the highest-bit and low-bit respectively. Hardware shift operations can efficiently execute the division and multiplication by $2^\Delta$. And the $\mathbf{z}_h$ is 0 for the weight quantization in this paper. The gradient formulation of *Double Rounding* for one-shot joint training is represented as follows:

$$\frac{\partial \widehat{Y}}{\partial \mathbf{s}_h} \simeq \begin{cases} \left\lfloor \frac{Y - \mathbf{z}_h}{\mathbf{s}_h} \right\rceil - \frac{Y - \mathbf{z}_h}{\mathbf{s}_h} & if\, n < \frac{Y - \mathbf{z}_h}{\mathbf{s}_h} < p, \\ n \quad or \quad p & otherwise. \end{cases} \tag{13}$$

$$\frac{\partial \widehat{Y}}{\partial \mathbf{z}_h} \simeq \begin{cases} 0 & if\, n < \frac{Y - \mathbf{z}_h}{\mathbf{s}_h} < p, \\ 1 & otherwise. \end{cases} \tag{14}$$

where $n$ and $p$ denote the lower and upper bounds of the integer range $[N_{min}, N_{max}]$ for quantizing the weights or activations respectively. $Y$ represents the FP32 weights or activations, and $\widehat{Y}$ represents the dequantized weights or activations. Unlike weights, activation quantization *scale* and *zero-point* of different precisions are learned independently. However, the gradient formulation is the same.

### A.5 ALGORITHMS

This section provides the algorithm implementations of multi-precision, one-shot mixed-precision joint training, and the search stage of SubNets.

#### A.5.1 MULTI-PRECISION JOINT TRAINING

The multi-precision model with different quantization precisions shares the same model weight(*e.g.*, the highest-bit) during joint training. In conventional multi-precision, the shared weight (*e.g.*, multi-precision model) computes $n$ forward processes at each training iteration, where $n$ is the number of candidate bit-widths. Then, all attained losses of different precisions perform an accumulation, and update the parameters accordingly. For specific implementation details please refer to Algorithm A.1.

However, we find that if separate precision loss and parameter updates are performed directly after calculating a precision at each forward process, it will lead to difficulty convergence during training

---

**Algorithm A.1** Conventional Multi-precision training approach

---

**Require:** Candidate bit-widths set $b \in B$;
1: Initialize: Pretrained model $M$ with FP32 weights $W$, the quantization scales $\mathbf{s}$ including of weights $\mathbf{s}_w$ and activations $\mathbf{s}_x$, BatchNorm layers:$\{BN\}_{b=1}^{n}$, optimizer:$optim(W, \mathbf{s}, wd)$, learning rate: $\lambda$, $wd$: weight decay, $CE$: CrossEntropyLoss, $D_{train}$: training dataset;
2: For one epoch:
3: Sample mini-batch data $(\mathbf{x}, \mathbf{y}) \in \{D_{train}\}$
4: **for** $b$ in $B$ **do**
5: $\quad forward(M, \mathbf{x}, \mathbf{y}, b)$:
6: $\quad$ **for** each quantization layer **do**
7: $\quad\quad \widehat{W}^b = dequant(quant(W, \mathbf{s}_w^b))$
8: $\quad\quad \widehat{X}^b = dequant(quant(X, \mathbf{s}_x^b))$
9: $\quad\quad O^b = Conv(\widehat{W}^b, \widehat{X}^b)$
10: $\quad$ **end for**
11: $\quad \mathbf{o}^b = FC(W, O^b)$
12: $\quad$ Update $BN^b$ layer
13: $\quad$ Compute loss: $\mathcal{L}^b = CE(\mathbf{o}^b, \mathbf{y})$
14: $\quad$ Compute gradients: $\mathcal{L}^b.backward()$
15: **end for**
16: Update weights and scales: $optim.step(\lambda)$
17: Clear gradient: $optim.zero\_grad()$;
**Note** that $n$ and $L$ represent the number of candidate bit-widths and model layers respectively.

---

or suboptimal accuracy. In other words, the varying gradient magnitudes of quantization scales of different precisions make it hard to attain stable convergence during joint training. To address this issue, we introduce an adaptive approach (*e.g.*, Adaptive Learning Rate Scaling, ALRS) to alter the learning rate for different precisions during training, aiming to achieve a consistent update pace. This method allows us to directly update the shared parameters after calculating the loss after every forward. We update both the weight parameters and quantization parameters simultaneously using dual optimizers. We also set the weight-decay of the quantization scales to 0 to achieve more stable convergence. For specific implementation details, please refer to Algorithm A.2.

---

**Algorithm A.2** Our Multi-precision training approach

---

**Require:** Candidate bit-widths set $b \in B$
1: Initialize: Pretrained model $M$ with FP32 weights $W$, the quantization scales $\mathbf{s}$ including of weights $\mathbf{s}_w$ and activations $\mathbf{s}_x$, BatchNorm layers: $\{BN\}_{b=1}^{n}$, optimizers: $optim_1(W, wd)$, $optim_2(\mathbf{s}, wd = 0)$, learning rate: $\lambda$, $wd$: weight decay, $CE$: CrossEntropyLoss, $D_{train}$: training dataset;
2: For every epoch:
3: Sample mini-batch data $(\mathbf{x}, \mathbf{y}) \in \{D_{train}\}$
4: **for** $b$ in $B$ **do**
5: $\quad forward(M, \mathbf{x}, \mathbf{y}, b)$:
6: $\quad$ **for** each quantization layer **do**
7: $\quad\quad \widehat{W}^b = dequant(quant(W, \mathbf{s}_w^b))$
8: $\quad\quad \widehat{X}^b = dequant(quant(X, \mathbf{s}_x^b))$
9: $\quad\quad O^b = Conv(\widehat{W}^b, \widehat{X}^b)$
10: $\quad$ **end for**
11: $\quad \mathbf{o}^b = FC(W, O^b)$
12: $\quad$ Update $BN^b$ layer
13: $\quad$ Compute loss: $\mathcal{L}^b = CE(\mathbf{o}^b, \mathbf{y})$
14: $\quad$ Compute gradients: $\mathcal{L}^b.backward()$
15: $\quad$ Compute learning rate: $\lambda_b$ $\quad$ # please see formula (6) of the main paper
16: $\quad$ Update weights and quantization scales: $optim_1.step(\lambda)$; $optim_2.step(\lambda_b)$
17: $\quad$ Clear gradient: $optim_1.zero\_grad()$; $optim_2.zero\_grad()$
18: **end for**
**Note** that $n$ and $L$ represent the number of candidate bit-widths and model layers respectively.

---

### A.5.2 ONE-SHOT JOINT TRAINING FOR MIXED PRECISION SUPERNET

Unlike multi-precision joint quantization, the bit-switching of mixed-precision training is more complicated. In multi-precision training, the bit-widths calculated in each iteration are fixed, *e.g.*, {8,6,4,2}-bit. In mixed-precision training, the bit-widths of different layers are not fixed in each iteration, *e.g.* {8,random-bit,2}-bit, where "random-bit" is any bits of *e.g.*, {7,6,5,4,3,2}-bit, similar to the *sandwich* strategy of (Yu et al., 2018). Therefore, mixed precision training often requires more training epochs to reach convergence compared to multi-precision training. Bit-mixer (Bulat & Tzimiropoulos, 2021) conducts the same probability of selecting bit-width for different layers. However, we take the sensitivity of each layer into consideration which uses sensitivity (*e.g.* Hessian Matrix Trace (Dong et al., 2020)) as a metric to identify the selection probability of different layers. For more sensitive layers, preference is given to higher-bit widths, and vice versa. We refer to this training strategy as a Hessian-Aware Stochastic Bit-switching (HASB) strategy for optimizing one-shot mixed-precision SuperNet. Specific implementation details can be found in Algorithm A.3. In additionally, unlike multi-precision joint training, the BN layers are replaced by TBN (Transitional Batch-Norm) (Bulat & Tzimiropoulos, 2021), which compensates for the distribution shift between adjacent layers that are quantized to different bit-widths. To achieve the best convergence effect, we propose that the threshold of bit-switching (*i.e.*, $\sigma$) also increases as the epoch increases.

---

**Algorithm A.3** Our one-shot Mixed-precision SuperNet training approach

---

**Require:** Candidate bit-widths set $b \in B$, the HMT of different layers of FP32 model: $t_l \in \{T\}_{l=1}^{L}$, average HMT: $t_m = \frac{\sum_{l=1}^{L} t_l}{L}$;

1: Initialize: Pretrained model $M$ with FP32 weights $W$, the quantization scales $\mathbf{s}$ including of weights $\mathbf{s}_w$ and activations $\mathbf{s}_x$, BatchNorm layers:$\{BN\}_{b=1}^{n^2}$, the threshold of bit-switching:$\sigma$, optimizer:$optim(W, \mathbf{s}, wd)$, learning rate: $\lambda$, $wd$: weight decay, $CE$: CrossEntropyLoss, $D_{train}$: training dataset;
2: For one epoch:
3:   Attain the threshold of bit-switching: $\sigma = \sigma \times \frac{epoch+1}{total\_epochs}$
4:   Sample mini-batch data $(\mathbf{x}, \mathbf{y}) \in \{D_{train}\}$
5:   **for** $b$ in $B$ **do**
6:     $forward(M, \mathbf{x}, \mathbf{y}, b, T, t_m)$:
7:     **for** each quantization layer **do**
8:       Sample $r \sim U[0, 1]$;
9:       **if** $r < \sigma$ **then**
10:         $b = Roulette(B, t_l, t_m)$      # Please refer to Algorithm 1 of the main paper
11:       **end if**
12:       $\widehat{W}^b = dequant(quant(W, \mathbf{s}_w^b))$
13:       $\widehat{X}^b = dequant(quant(X, \mathbf{s}_x^b))$
14:       $O^b = Conv(\widehat{W}^b, \widehat{X}^b)$
15:     **end for**
16:     $\mathbf{o}^b = FC(W, O^b)$
17:     Update $BN^b$ layer
18:     Compute loss: $\mathcal{L}^b = CE(\mathbf{o}^b, \mathbf{y})$
19:     Compute gradients: $\mathcal{L}^b.backward()$
20:     Update weights and scales: $optim.step(\lambda)$
21:     Clear gradient: $optim.zero\_grad()$;
22: **end for**

**Note** that $n$ and $L$ represent the number of candidate bit-widths and model layers respectively.

---

### A.5.3 EFFICIENT ONE-SHOT SEARCHING FOR MIXED PRECISION SUPERNET

After training the mixed-precision SuperNet, the next step is to select the appropriate optimal Sub-Nets based on conditions, such as model parameters, latency, and FLOPs, for actual deployment and inference. To achieve optimal allocations for candidate bit-width under given conditions, we employ the Iterative Integer Linear Programming (ILP) approach. Since each ILP run can only provide one solution, we obtain multiple solutions by altering the values of different average bit widths. Specifically, given a trained SuperNet (*e.g.*, RestNet18), it takes less than two minutes to solve candidate SubNets. It can be implemented through the Python `PULP` package. Finally, these searched SubNets only need inference to attain final accuracy, which needs a few hours. This forms a Pareto

optimal frontier. From this frontier, we can select the appropriate subnet for deployment. Specific implementation details of the searching process by ILP can be found in Algorithm 2.

## A.6 THE GRADIENT STATISTICS OF LEARNABLE SCALE OF QUANTIZATION

In this section, we analyze the changes in gradients of the learnable scale for different models during the training process. Figure 7 and Figure 8 display the gradient statistical results for ResNet20 on CIFAR-10. Similarly, Figure 9 and Figure 10 show the gradient statistical results for ResNet18 on ImageNet-1K, and Figure 11 and Figure 12 present the gradient statistical results for ResNet50 on ImageNet-1K. These figures reveal a similarity in the range of gradient changes between higher-bit quantization and 2-bit quantization. Notably, they illustrate that the value range of 2-bit quantization is noticeably an order of magnitude higher than the value ranges of higher-bit quantization.

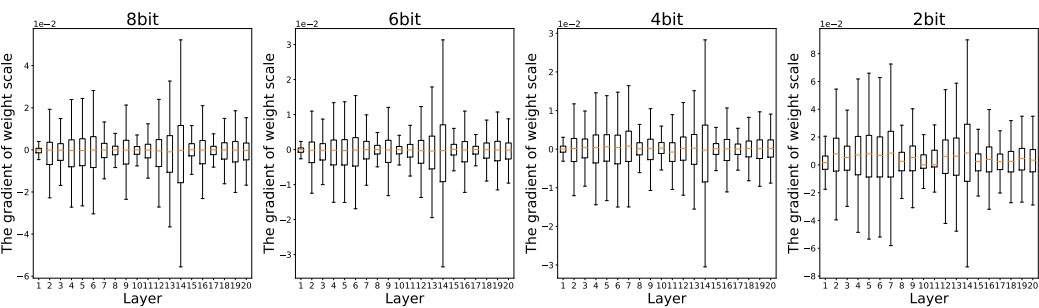

Figure 7: The scale gradient statistics of weight of ResNet20 on CIFAR-10 dataset. Note that the outliers are removed for exhibition.

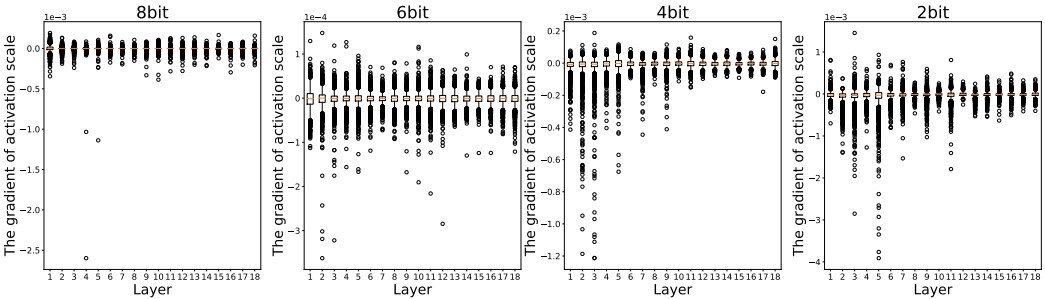

Figure 8: The scale gradient statistics of activation of ResNet20 on CIFAR-10 dataset. Note that the first and last layers are not quantized.

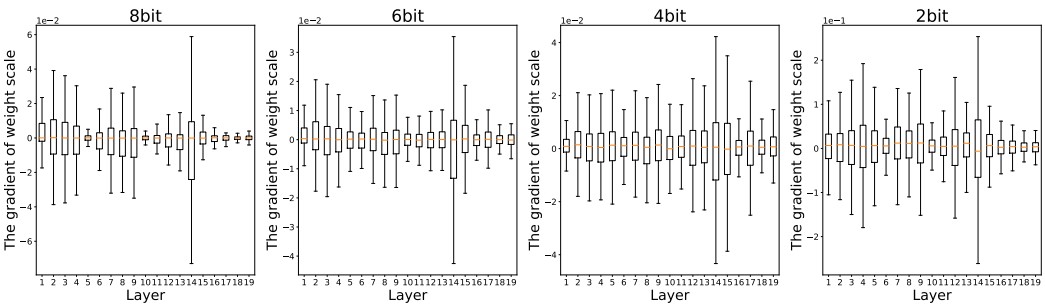

Figure 9: The scale gradient statistics of weight of ResNet18 on ImageNet dataset. Note that the outliers are removed for exhibition.

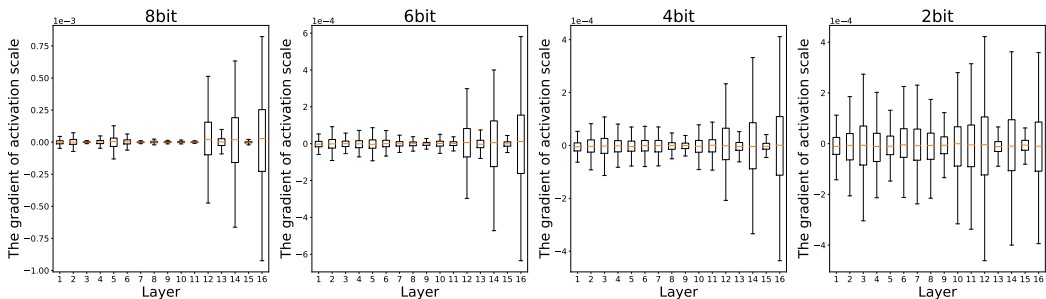

Figure 10: The scale gradient statistics of activation of ResNet18 on ImageNet dataset. Note that the outliers are removed for exhibition.

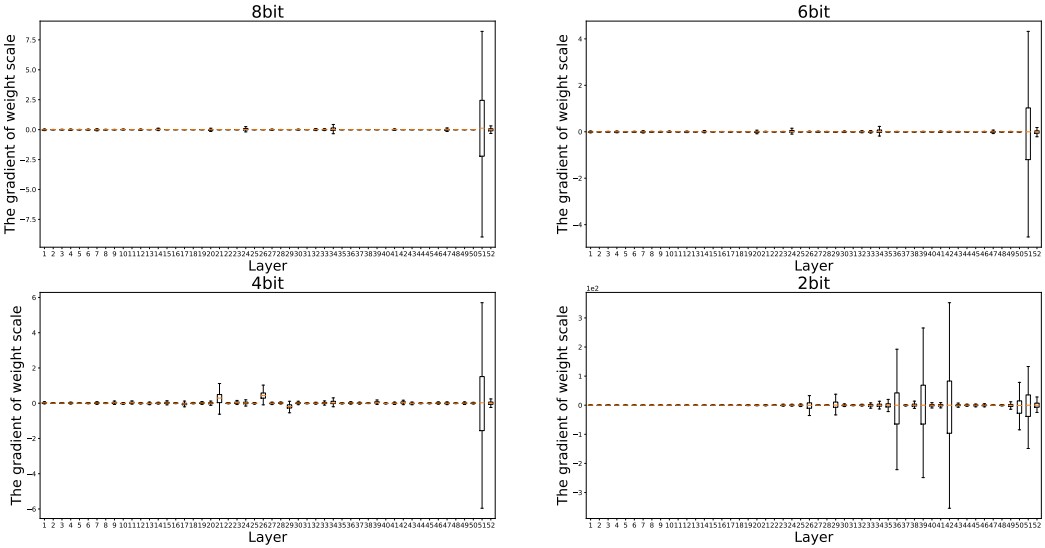

Figure 11: The scale gradient statistics of weight of ResNet50 on ImageNet dataset. Note that the outliers are removed for exhibition, and the first and last layers are not quantized.

### A.7 MIXED-PRECISION BIT ALLOCATION IN DIFFERENT LAYERS

We also provide the searched per-layer bit-width results for the one-shot mixed-precision experiments on ResNet-18, ResNet-50 and MobileNet-V2. These results can be found in Figure 13, Figure 14 and Figure 15. As shown in Figures 13, 14 and 15, for the mixed-precision bit-width distributions learned using the HASB technique, lower given average bit-widths result in more high-bit allocations being directed towards sensitive regions, which aligns closely with the corresponding HMT curve trends. In contrast, the bit-width distributions learned without the HASB technique tend to exhibit more randomness and deviate from the HMT curve. These results further validate the effectiveness of the proposed HASB technique.

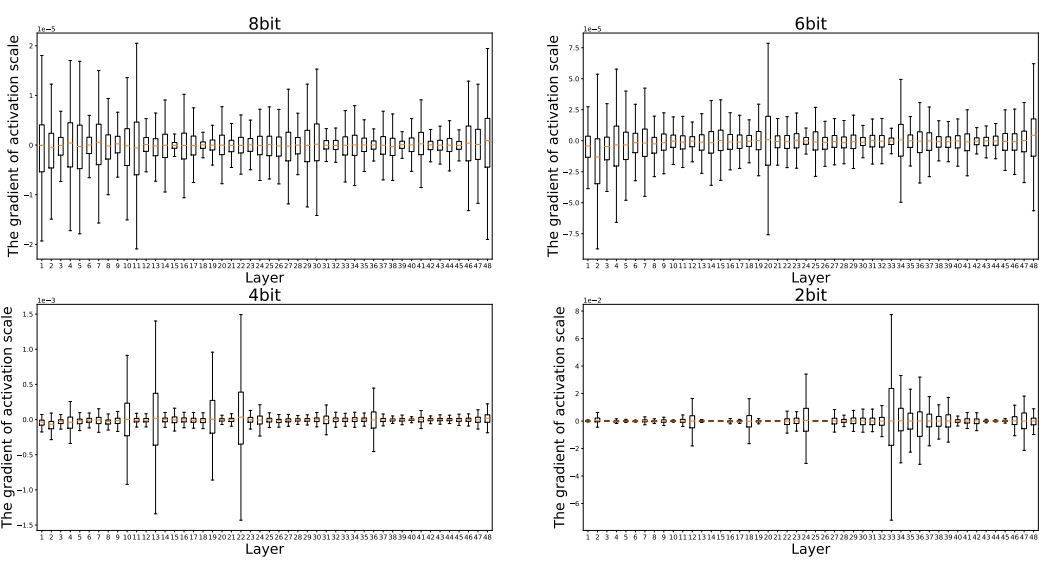

Figure 12: The scale gradient statistics of activation of ResNet50 on ImageNet dataset. Note that the outliers are removed for exhibition.

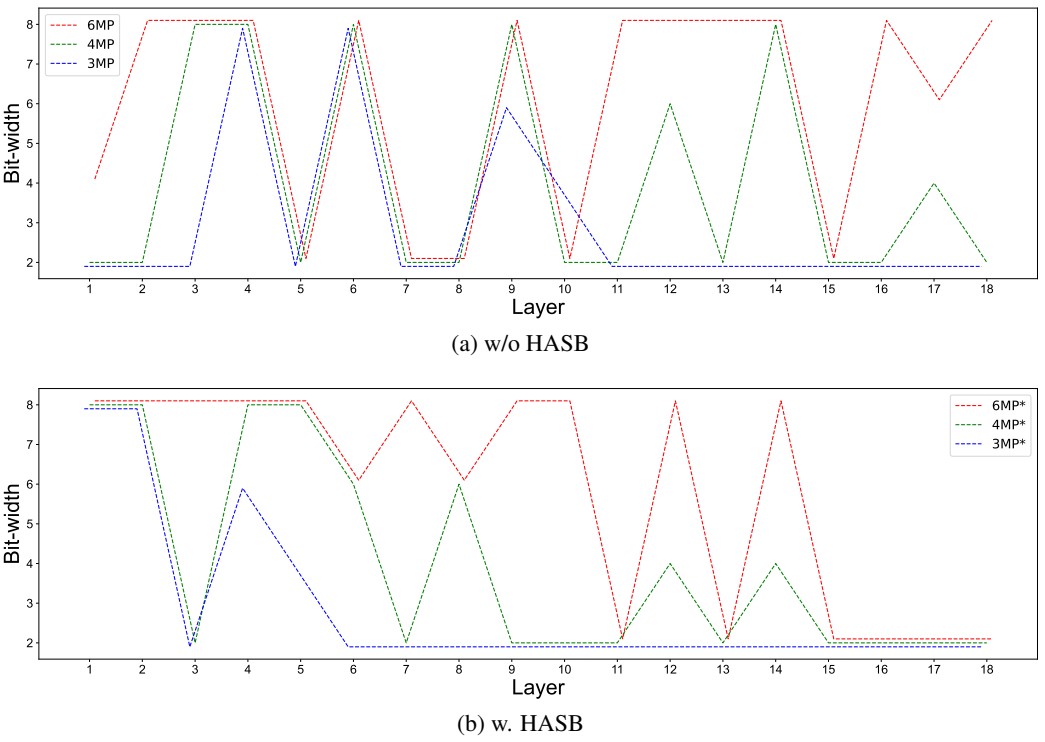

Figure 13: Layer-Wise Bit-Widths Allocation of Mixed-Precision in ResNet18. (a) Without HASB, bit-widths shows a more random allocation and is less aligned with the sensitivity trends. (b) With HASB, bit-widths are effectively allocated to sensitive layers based on HMT curves.

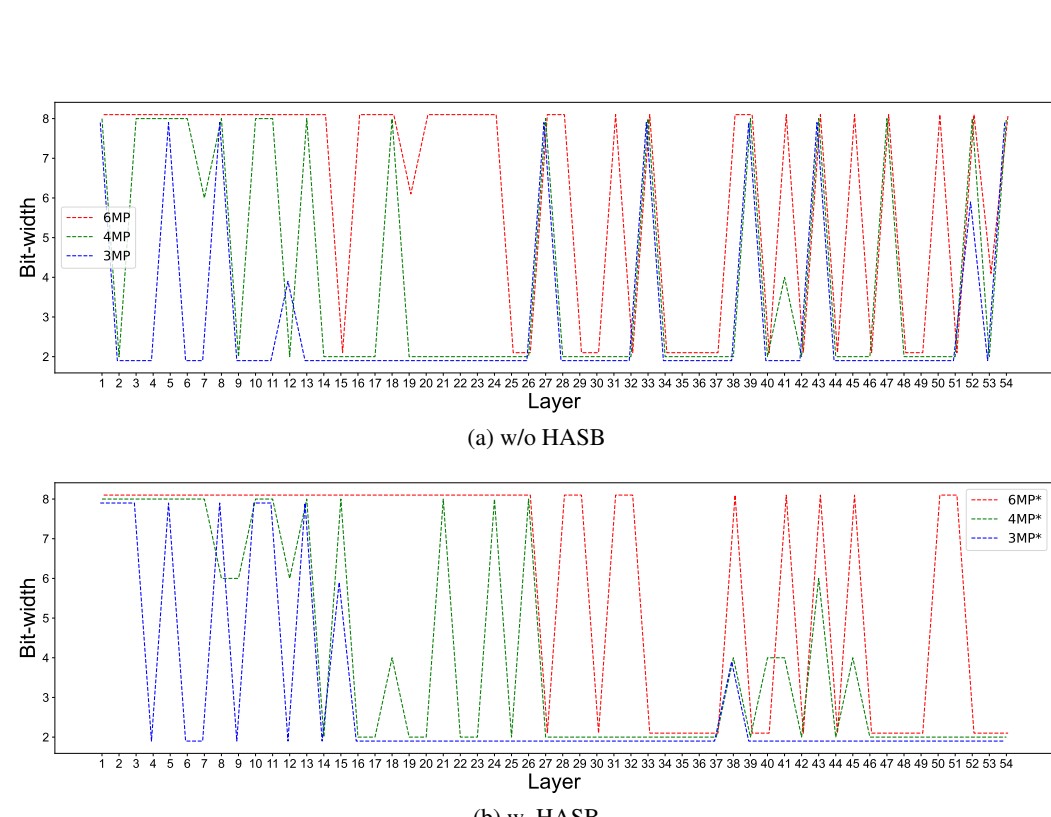

(a) w/o HASB

(b) w. HASB

Figure 14: Layer-Wise Bit-Widths Allocation of Mixed-Precision in ResNet50. (a) Without HASB, bit-widths shows a more random allocation and is less aligned with the sensitivity trends. (b) With HASB, bit-widths are effectively allocated to sensitive layers based on HMT curves.

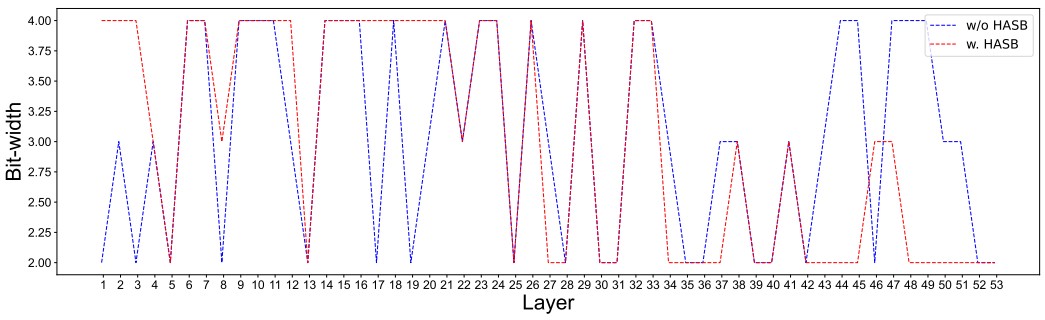

Figure 15: Layer-Wise Bit-Widths Allocation of Mixed-Precision (3MP) in MobileNet-v2.

