# OpenReview forum: "Nearly Lossless Adaptive Bit Switching"
_ICLR.cc/2025/Conference — ICLR 2025 Conference Withdrawn Submission_

### Official Review · Reviewer_Psfe · 2024-11-01

**Soundness:** 4
**Presentation:** 3
**Contribution:** 3
**Rating:** 8
**Confidence:** 5

**Summary:**

This paper presents an approach to address the challenges of multi-precision and mixed-precision joint training in deep neural network quantization. The authors introduce a technique called double rounding quantization, which allows for the storage of a single integer weight representation while enabling adaptive precision switching with minimal accuracy loss. This method utilizes two rounding operations, effectively embedding lower-precision weights within higher-precision weights, thus reducing storage requirements. The paper further introduces an adaptive learning rate scaling (ALRS) method designed to mitigate the competitive interference between high and low precisions during training. ALRS dynamically adjusts the learning rates for different precisions, trying to ensure consistent update steps of quantization scales and improving the accuracy of low-precision models without sacrificing the performance of high-precision models.  Furthermore, the authors propose a hessian-aware stochastic bit-switching (HASB) strategy for one-shot mixed-precision training. HASB leverages the hessian matrix trace as a sensitivity metric to determine the probability of bit-width selection for each layer, prioritizing higher bit-widths for more sensitive layers.

**Strengths:**

1. Findings: Joint optimization of quantized network for multiple precision results to competition between high and low precisions. However, the reason was unclear in existing works. This paper point out that the reason of the competition comes from the inconsistent magnitudes of the quantization scale’s gradients between high precision and low precision.
2. Strong multi precision result: The results in Table2 and Table6 show that, in multi precision, double rounding and ALRS each show effective result compared to existing work. Although the effect of double rounding only is not directly revealed in the paper, it can be sufficiently inferred by comparing the two tables.

**Weaknesses:**

Please see the questions below.

**Questions:**

1. Is there any reason the MobileNet-v2 result is not included on mixed precision results? Many existing works show that the quantization methods that show good result on ResNets cannot applicable to MobileNets which is hard to quantize.
2. This reviewer suggests proposing searched per-layer bit-width result for the one-shot mixed precision result (of proposed models, e.g, ResNet-18 and ResNet-50). This would increase the understanding of effect of HASB with hessian trace by comparing it to a baseline without using HASB.
3. On Figure 4(d), it seems the results having same average bit-width is showing large accuracy differences. Does it mean that the same mixed-precision budget can show different per-layer bit-width results? If so, this results show that HASB cannot always show optimal result. Maybe the authors can analyze some specific cases where the same average bit-width produces significantly different accuracies, explaining what causes these differences and how this relates to the effectiveness of HASB.
4. Algorithm: Lack of explanation of several terms in Algorithm2 makes bad readability. Please define 'pulp' and 'ideas' in the context of Algorithm2, either within the algorithm description or in an accompanying glossary of terms.

---

> ### Author Response · Authors · 2024-11-19
>
> Dear Reviewer Psfe,
>
> We greatly appreciate your time and efforts for this paper's improvement and here are our responses:
>
> **Q1:** Is there any reason the MobileNet-v2 result is not included on mixed precision results?
> **A1:** Thank you for your valuable suggestion.
> In fact, we have also conducted experiments on MobileNet-v2 for mixed precision, and the results are provided in the following table. These results demonstrate that our method is also effective on MobileNet-v2. The reason we initially omitted these results is that all of the comparison methods (i.e., Bit-Mixer, ABN, MultiQuant, EQ-Net) do not report results and fail to achieve convergence on MobileNet-v2 under the {4,3,2} configuration. Including these comparisons further highlights the robustness and effectiveness of our proposed method.
> | Model       | Dataset  | KD | Training | Searching | Fine-tune | Epoch | w4a4  | w3a3  | w2a2  | 3MP   | FP    |
> |:-----------:|:--------:|:---:|:--------:|:---------:|:---------:|:-----:|:-----:|:-----:|:-----:|:-----:|:-----:|
> | MobileNetV2 | Cifar10  | ×  | HASB     | ILP       | w/o       | 90    | 93.21 | 92.95 | 87.57 | 93.34 | 93.47 |
> |             |          | √  | HASB     | ILP       | w/o       | 90    | 93.46 | 93.22 | 88.24 | 93.52 | 93.47 |
> | MobileNetV2 | ImageNet | ×  | HASB     | ILP       | w/o       | 90    | 69.32 | 68.47 | 52.14 | 69.47 | 71.14 |
> |             |          | √  | HASB     | ILP       | w/o       | 90    | 69.86 | 68.55 | 56.19 | 69.89 | 71.14 |
>
> **Q2:** This reviewer suggests proposing a searched per-layer bit-width result for the one-shot mixed precision result.
> **A2:** Thank you for your constructive suggestion.
> We have provided the searched per-layer bit-width results for the one-shot mixed-precision experiments on ResNet-18 and ResNet-50. These results can be found in **Fig. 13 and 14**, which **we have added to the updated appendix of the revised manuscript.** As shown in Fig. 13 and 14, for the mixed-precision bit-width distributions learned using the HASB technique, lower given average bit-widths result in more high-bit allocations being directed towards sensitive regions, which aligns closely with the corresponding HMT curve trends. In contrast, the bit-width distributions learned without the HASB technique tend to exhibit more randomness and deviate from the HMT curve. These results further validate the effectiveness of the proposed HASB technique.
>
> **Q3:** Explain what causes significantly different accuracies under the same average bit-width and how this relates to the effectiveness of HASB.
> **A3:** Thanks for your valuable comments.
> The reasons for the observed accuracy differences with the same average bit-width are as follows:
> - **ILP Approximation**: The Integer Linear Programming (ILP) solver provides an approximate solution rather than an exact theoretical optimum. This discrepancy is particularly pronounced at lower average bit-widths, where the constraints are tighter. Improving the ILP search strategy in future work may help reduce accuracy differences.
> - **Constraint Factors in ILP**: The accuracy of the ILP solution depends on the constraint factors provided during the search. In our work, we primarily consider the Hessian Matrix Trace (HMT) of different layers and the average bit-width (equivalent to parameter count), as described in Algorithm 2. Including additional constraints, such as FLOPs, may lead to more precise solutions and higher accuracy.
> - **Impact of HASB on SuperNet Training**: While HASB plays a role during mixed-precision training, the quality of the trained SuperNet directly affects the ILP solution. Enhancing the HASB training strategy (e.g., refining the sensitivity threshold by using a top-k approach, where the threshold is set as the mean HMT of the top-k most sensitive layers) could further improve mixed-precision performance and reduce accuracy differences.
> We hope this explanation clarifies the underlying causes of these differences and demonstrates potential avenues for improvement. Thank you again for your insightful feedback.
>
> **Q4:** Lack of explanation of several terms in Algorithm2 makes bad readability.
> **A4:** Thank you for your valuable suggestion.
> The term "pulp" in Algorithm 2 refers to the Python library PuLP, which is used for linear programming and integer optimization. In our context, it is employed to define and solve ILP problems for finding the optimal bit-width configuration.
> Regarding "ideas," we believe you are referring to the term "idenx," which is indeed a typo. It should be "index," representing the position of the maximum bit-width value in the candidate bit-widths.
> We have provided detailed explanations of the terms in Algorithm 2 to enhance readability. Thanks very much for bringing this to our attention.
>
> Sincerely,
>
> Paper #5733 Authors

---

> ### Author Response · Authors · 2024-11-21
> **Looking forward to your feedback**
>
> Dear Reviewer Psfe,
>
> With the discussion phase nearing the end, we would like to know whether the responses have addressed your concerns.
>
> Should this be the case, we are encouraged that you raise the final rating to reflect this.
>
> If there are any remaining concerns, please let us know. We are more than willing to engage in further discussion and address any remaining concerns to the best of our abilities.
>
> We are looking forward to your reply. Thank you for your efforts in this paper.
>
> Best regards,
>
> Paper #5733 Authors

---

> > ### Comment · Reviewer_Psfe · 2024-11-25
> >
> > Thanks a lot for answering the questions.
> > Below are additional questions for your answers.
> >
> > A1: Beyond multi precision works, existing works only targeting mixed precision are failing to converge on MobileNet-v2 when including 2-bit as bit-width candidate. Do the authors have any reason for this phenomenon? Maybe authors can also propose the searched per-layer bit-width result for the one-shot mixed precision result in A1.
> >
> > A3: Then it sounds like the effect of HASB cannot be the best solution compared to others in Table 3 in different settings. Figure 5 shows the possibility of inferiority when the bit configuration is changed or target average bit-width (MP) is changed. (3MP on {4,3,2}-bit seems the best improvement scenario to show the effect of HASB while others are in mysterious state.)

---

> ### Author Response · Authors · 2024-11-25
>
> Dear Reviewer Psfe,
>
> Thank you for your constructive feedback and for acknowledging the efforts we have made to address your previous concerns.
>
> Re-A1:
> Under the one-shot mixed-precision joint optimization framework, the conflict or competition between 2-bit and higher-bit precisions becomes more pronounced in networks like MobileNet-v2, which are inherently more challenging to quantize. Our method effectively addresses this issue by employing double rounding and HASB techniques to reduce the precision loss when switching from higher-bit to lower-bit representations.
> Additionally, we have included the searched per-layer bit-width results for the one-shot mixed-precision experiments {4,3,2}-bits on MobileNet-v2 in Fig. 15 of the updated appendix in the revised manuscript.
>
> Re-A3:
> As observed in Table 3, our method achieves comparable results for the same bit-width settings (w4a4, w3a3, w2a2) with **fewer training epochs** (90 vs.160 epochs), and achieves the best results under mixed precision (3MP). Previously, we have conducted experiments on multi-precision settings, which show that more epoch training significantly enhances model accuracy, as demonstrated in the table below.
> | Model    | Dataset  | Method  | Epoch | w4a4  | w3a3  | w2a2  | FP    |
> |----------|----------|---------|-------|-------|-------|-------|-------|
> | ResNet50 | ImageNet | AdaBits | 150   | 76.10 | 75.80 | 73.20 | 75.00 |
> |          |          | Ours*   | 90    | 76.42 | 75.82 | 73.28 | 76.13 |
> |          |          | Ours*   | 150   | 76.65 | 75.96 | 73.40 | 76.13 |
> |          |          | Ours*   | 300   | 76.78 | 76.05 | 73.50 | 76.13 |
>
> Moreover, compared to previous methods, our proposed HASB technique not only improves model accuracy but also enables the super-net obtained after training to directly achieve the optimal candidate bit-width configuration through ILP search **without additional fine-tuning or re-training**. This considerably reduces the cost of model optimization.
>
> As illustrated in Fig.5, the potential degradation under different bit-width configurations can be attributed to other influencing factors, as summarized in A3. And, we have updated Fig. 5 (b) in the revised version to include results with the Top-k strategy applied. As we can see, the advantage of HASB is more obvious with the favorable strategy.
>
> Finally, if you feel that your further concerns are addressed, we would appreciate updating your evaluation to reflect that. Thanks very much for your time and efforts for this paper. Looking forward to your reply.
>
> Sincerely
>
> Paper #5733 Authors

---

> ### Comment · Reviewer_Psfe · 2024-12-02
>
> Thanks a lot for answering my questions.
> Still my concern is not resolved in terms of mixed precision (possibility of inferiority other than 3MP).
>
> However, I think this paper has more contributions on multi-precision results and possibility of 2-bit as bit-width candidate of mixed precision on MobileNet.
> I hope you release your anonymous code on the final version of the paper and including our discussions on appendix.

---

> > ### Author Response · Authors · 2024-12-02
> >
> > Dear Reviewer Psfe,
> >
> > Thanks very much for your construction feedback and we are grateful for your recognition of our work.
> >
> > We promise that we would release the anonymous code on the final version of the paper and including the discussions of the reviewers on appendix. Thanks again for your time and attention on this paper.
> >
> > Best regards,
> >
> > Paper #5733 Authors

---

### Official Review · Reviewer_X94Z · 2024-11-04

**Soundness:** 3
**Presentation:** 2
**Contribution:** 2
**Rating:** 5
**Confidence:** 4

**Summary:**

The paper introduces a quantization method aimed at improving DNN compression with flexible, precision-switching capabilities that reduce storage and accuracy trade-offs. Traditional Quantization-Aware Training uses fixed, uniform bit-widths, but this approach instead uses three key techniques to optimize multi-precision performance. First, Double Rounding applies a double rounding operation that minimizes accuracy loss when switching between precisions, storing only the highest integer precision and eliminating the need for full-precision storage. Second, Adaptive Learning Rate Scaling dynamically adjusts learning rates across bit-widths, reducing gradient interference and balancing convergence between high and low precisions during one-shot joint training. Third, Hessian-Aware Stochastic Bit-switching uses layer sensitivity, measured by the Hessian Matrix Trace, to assign higher bit-widths to more sensitive layers and lower ones to less sensitive layers, optimizing precision across the network. Together, these methods demonstrate competitive performance in ImageNet-1K classification, detection, and segmentation tasks, surpassing current multi-precision and mixed-precision quantization approaches.

**Strengths:**

- The paper is generally well-written, with clear explanations of concepts and effective use of figures to illustrate key ideas, such as the interference issue during multi-precision training and the Double Rounding operation.
- The proposed Double Rounding method offers a solution for precision-switching without significant storage requirements or accuracy loss.

**Weaknesses:**

- The paper’s experiments are limited to CNNs on ImageNet for classification, lacking a broader evaluation across different architectures, such as transformers, which are essential for both CV and NLP tasks. For NLP, the use of TinyLlama as a baseline is insufficient, as it is not a widely recognized standard for large language models. A more rigorous evaluation on established baselines, such as LLama-3, would provide a stronger and more credible assessment of the method’s effectiveness across diverse model types and tasks.
- While the authors demonstrate results on ImageNet-1K for classification, the experiments lack evaluation on more complex or fine-grained tasks (e.g., instance segmentation, object detection on datasets beyond ImageNet). These tasks often pose unique quantization challenges due to greater sensitivity to detail and precision.
- The paper could benefit from more comprehensive ablation studies, especially on the impact of each component (Double Rounding, ALRS, and HASB) independently and in combination.

**Questions:**

- Given the paper's focus on mobile and edge scenarios, could the authors quantify the memory and power consumption improvements (if any) compared to traditional quantization methods?
- While ALRS claims to improve convergence consistency across bit-widths, could the authors provide a theoretical explanation or further empirical analysis to support this?

---

> ### Author Response · Authors · 2024-11-19
>
> Dear Reviewer X94Z,
>
> We greatly appreciate your time and efforts for this paper's improvement and here are our responses:
>
> **W1:** The paper’s experiments are limited.
> **A1:**  Thanks for your valuable comments.
> Due to time and computational resource constraints, we have tried our best to add experiments on ViT-B/16 and ViT-B/32 models for visual classification tasks, as well as Llama-3-8B for zero-shot inference in NLP. However, the experiments are still not finished and we will include the results once they are done.
>
> **W2:** Lack of evaluation on instance segmentation and object detection.
> **A2:** Thanks for your comments.
> We have validated the effectiveness of our method on object detection and instance segmentation tasks using the Mask-RCNN model on the COCO dataset, as shown in Table 4 of the paper.
>
> **W3:** The paper could benefit from more comprehensive ablation studies.
> **A3** Thanks for your comments.
> Firstly, Double Rounding is the core method proposed in our work and serves as the baseline for our experiments. For results without Double Rounding, please refer to the results of MultiQuant and Bit-Mixer in Tables 1 and 2, respectively. As highlighted by Reviewer Psfe in "Strengths: 2," a comparison of these tables sufficiently demonstrates the effectiveness of Double Rounding. Secondly, the results of ALRS, both when combined with Double Rounding and when used independently, can be found in Tables 6 and 7. Lastly, for the experimental results of HASB, please refer to Figure 5 in the paper.
>
> **Q1:** Could the authors quantify the memory and power consumption improvements compared to traditional quantization methods?
> **A4:** Thanks for your valuable question.
> Firstly, the memory and power consumption during inference can be approximately quantified using the "average bit-width × model FLOPs" metric after quantization. This provides an indirect estimate of the actual memory and power requirements of the model during deployment since the proposed Double Rounding method is hardware-friendly and capable of reducing power consumption during inference. Specifically, the division by $2^Δ$ operations can be efficiently achieved by 1 << Δ on the mobile or edge scenarios. Combined with the nearly lossless bit-switching capability, our method allows adaptive model precision switching during inference based on task complexity, thereby lowering power consumption and accelerating the inference process.
>
> **Q2:** Could the authors provide a theoretical explanation or further empirical analysis of ALRS?
> **A5:** Thanks for your valuable question. We attempt to provide an intuitive, though not fully rigorous, theoretical explanation of how ALRS adapts the quantization scale and model weight parameters.
> ALRS is specifically applied to the quantization scale, as this parameter is highly sensitive during quantization-aware training and plays a crucial role in determining the final convergence performance. In our multi-precision or mixed-precision scenarios, the model weights are shared across different precisions. Directly scaling the shared weights would disrupt the convergence of other precisions due to the strong competition between them. By applying the ALRS strategy to the quantization scales across different precisions (including smaller bit-widths), it indirectly adjusts the effective scaling of the weights for those bit-widths. The quantization process can be formulated as:
> $W_i = Round(W_{i-1} / s_i ) * s_i + \Delta_w(x) * lr$,
> $s_i = s_{i-1} + \Delta_s(x) * ALRS $,
> where the $\Delta_w$ and $\Delta_s$ denote the gradients of the $W_{i-1}$ and $s_{i-1}$ respectively, and $ALRS$ denotes the adaptive learning rate of the quantization scaling factors and $lr$ denotes the learning rate of the weights.
> This adaptive adjustment helps maintain consistent convergence across bit-widths, minimizing the training competition and improving performance. We hope this explanation clarifies the reasoning behind ALRS and its effectiveness in addressing convergence challenges in multi-precision training.
>
> Sincerely,
>
> Paper #5733 Authors

---

> ### Author Response · Authors · 2024-12-02
> **Looking forward to your feedback**
>
> Dear Reviewer X94Z,
>
> We apologize for the repeated reminder. It has been **12 days** since we submitted our responses, but we have not yet received your feedback. We simply want to ensure that we have fully addressed all your concerns.
>
> If there are any remaining points that require further clarification, please rest assured that we are committed to providing detailed answers to all your inquiries and addressing any concerns you may have. We value clear and open communication and will make every effort to ensure that all aspects of the matter are fully explained to your satisfaction.
>
> Thanks very much for your time and effort on this paper.
>
> Best regards,
>
> Paper #5733 Authors

---

> > ### Author Response · Authors · 2024-12-02
> > **Looking forward to your feedback**
> >
> > Dear Reviewer X94Z,
> >
> > With the discussion phase nearing the end, we would like to know whether the responses have addressed your concerns.
> >
> > Should this be the case, we are encouraged that you raise the final rating to reflect this.
> >
> > If there are any remaining concerns, please let us know. We are more than willing to engage in further discussion and address any remaining concerns to the best of our abilities.
> >
> > We are looking forward to your reply. Thank you for your efforts in this paper.
> >
> > Best regards,
> >
> > Paper #5733 Authors

---

### Official Review · Reviewer_qsNB · 2024-11-04

**Soundness:** 2
**Presentation:** 2
**Contribution:** 2
**Rating:** 5
**Confidence:** 4

**Summary:**

This paper proposes a bit-switching quantization method for multi-precision training. It employs a Double Rounding method that enables the storage of a shared 8-bit model instead of a full-precision one. To facilitate training convergence, the authors introduce the ALRS method to dynamically adjust the learning rate for each precision during training. The authors also apply the proposed bit-switching quantization to mixed-precision training for efficient subnet selection.

**Strengths:**

This paper is easy to follow, and the proposed approaches make sense.

**Weaknesses:**

1) The contribution of the Double Rounding technique does not appear to be significant. The first stage of DR seems aimed at avoiding the storage of a full-precision model during inference, but this concept is well-trodden ground. Similar approaches, such as starting with an 8-bit model in multi-precision or bit-sparse inference, have been extensively explored [1][2]. Considering that 8-bit quantization can preserve accuracy effectively, why not begin with an 8-bit model and perform subsequent processing?

2) The ALRS strategy appears straightforward and seems to have minor differences compared to training each precision separately. While the method works, my concern lies in the runtime memory overhead. ALRS may require keeping multiple copies of activations and gradients for each precision during training, potentially leading to increased memory usage. This could impact the scalability and efficiency of the approach, especially for large-scale models like LLMs.

3) The probability distributions used in Fig. 4(a) and Fig. 4(b) are not well motivated. For the {8, 6, 4, 2} configuration, it seems that the average number of bits for the insensitive and sensitive cases are 5 bits and 6 bits, respectively. Is this right? How can you achieve an average number of bits lower than 5, for example, 4 bits? What are the probability distributions when only three candidate bits are used, as in Table 3? As you recommend including 8-bit for training, why not evaluate {8,6,4,2} for the experiment in Table 3?

4) The accuracy of FP models is inconsistent in experiments, which is somewhat misleading. In Table 2, while the proposed method (ours*) achieves the highest accuracy, AdaBits experiences a much smaller relative accuracy drop.

5) It is unclear how knowledge distillation is performed in multi-precision training.

[1] Training for multi-resolution inference using reusable quantization terms. ASPLOS'21

[2] BitWave: Exploiting Column-Based Bit-Level Sparsity for Deep Learning Acceleration. HPCA'24

**Questions:**

Please see the weaknesses.

---

> ### Author Response · Authors · 2024-11-19
>
> Dear Reviewer qsNB,
>
> We greatly appreciate your time and efforts for this paper's improvement and here are our responses:
>
> **W1:** Why not begin with an 8-bit model and perform subsequent processing?
> **A1:**  Thank you for your valuable feedback.
> **Firstly**, we carefully reviewed the references you provided ([1] and [2]). [1] introduces a specialized hardware unit—the multi-resolution multiply-accumulator (mMAC), which processes different quantization levels by focusing only on significant power-of-two terms in the data. It also proposes Term Quantization (TQ), a method that quantizes weights or activation values by selecting a specific number of power-of-two terms. However, the TQ method has some key differences from our approach: (1) TQ fundamentally relies on a polynomial combination of power-of-two quantization, with a fixed quantization scale (i.e., sf, as seen in the TQ authors' [implementation](https://github.com/saizhang0218/Multi-resolution-Inference/blob/450d140fb8072b8c46b29ec7c0cda3cf5ea3ad8e/Term-quantization/models/quant_layer.py#L43)), which is similar to AdaBits in that the scale is not learnable. In contrast, our quantization scale is learnable, allowing the model to adapt optimally through training to different datasets and tasks. (2) TQ depends on specific hardware design (i.e., mMAC), whereas our method is designed for use on general-purpose hardware such as GPUs, CPUs, and FPGAs without requiring any modifications, making it more versatile. [2] introduces a new computing method called "bit-column-serial" along with a compatible architecture design, "BitWave." BitWave is designed to leverage structured bit-level sparsity and dynamic dataflow to reduce computation and memory usage. However, it is primarily focused on 8-bit Post-Training Quantization (PTQ) and does not align closely with our work on multi-bit Quantization-Aware Training (QAT) for joint quantization across multiple precisions. We have incorporated references [1] and [2] into the related work section of the revised manuscript.
> **Secondly**, our Double Rounding method is not only aimed at reducing storage but also enabling adaptive bit-switching during inference. This capability accelerates the inference process and reduces power consumption by adjusting the bit-width according to task requirements. Below, we outline some potential application scenarios to further illustrate the value of our approach:
> - **Dynamic Adaptation for Energy Efficiency**: In resource-constrained environments, our method can dynamically switch to lower bit-widths to save energy without significant accuracy loss.
> - **Latency-Sensitive Applications**: For applications that require low latency, such as real-time video processing, Double Rounding allows adaptive precision reduction to meet strict latency requirements.
>
> We hope this clarification and the potential applications help to better convey the unique contributions and significance of our approach.
>
> **W2:** The ALRS strategy may potentially lead to increased memory usage and impact scalability and efficiency.
> **A2:** Thank you for your kind comments.
> In fact, ALRS does not require storing multiple copies of activations or gradients for each precision during training. Instead, it only reads and scales the gradients at the point of loss computation, after which the gradients are immediately released. This approach minimizes memory usage, resulting in nearly identical memory consumption to training without ALRS. For further details, please refer to lines 505-526 in https://anonymous.4open.science/r/Double-Rounding-EF78/code/Multi_Precision/train.py in the code we provided. In addition, we provide the memory consumption results during training for different methods, as shown in the following table. We hope this explanation, along with the code reference, addresses your concern.
> | Model    | Dataset  | Method  | ALRS | Bit-widths    | BatchSize | Memory (G) |
> |----------|----------|---------|------|---------------|-----------|------------|
> | ResNet18 | ImageNet | w/o     | w/o  | FP32          | 256       | 7.88       |
> |          |          | LSQ     | w/o  | Separate-bit  | 256       | 10.01      |
> |          |          | Adabits | w/o  | {8,6,4,2}-bit | 256       | 10.02      |
> |          |          | Ours    | w/o  | {8,6,4,2}-bit | 256       | 10.03      |
> |          |          | Ours    | w    | {8,6,4,2}-bit | 256       | 10.09      |

---

> ### Author Response · Authors · 2024-11-19
>
> **W3:** The probability distributions used in Fig. 4(a) and Fig. 4(b) are not well motivated.
> **A3:** Thanks for your valuable comments.
> **Firstly**, the probability distributions used in Figures 4(a) and 4(b) are based on a straightforward weighted sampling approach. For insensitive layers, we perform uniform random sampling across the {8, 6, 4, 2} bit configuration, giving each candidate a probability density of 1/4. For sensitive layers, we assign probabilities in a 4:3:2:1 ratio for 8:6:4:2 bits, respectively, resulting in probability densities of 4/(4+3+2+1), 3/(4+3+2+1), 2/(4+3+2+1), and 1/(4+3+2+1).
> The expected bit-width, as you noted, is 5 bits for insensitive layers (0.25 * (2 + 4 + 6 + 8) = 5) and 6 bits for sensitive layers (0.1 * 2 + 0.2 * 4 + 0.3 * 6 + 0.4 * 8 = 6). To achieve a lower average bit-width, we could adjust the probabilities accordingly. For instance, for sensitive layers, we could set the probabilities to achieve a 4-bit average by replacing 0.25 with 0.2 in the uniform distribution. For insensitive layers, adjusting the weights to distributions such as {0.1, 0.1, 0.2, 0.4} or {0.1, 0.2, 0.1, 0.3} would also yield an average of 4-bit.
> **Secondly**, when only three candidate bits are used, such as {4, 3, 2}, we apply a similar approach: for insensitive layers, the probability for each bit is 1/3, while for sensitive layers, we use the ratio 4:3:2, resulting in probabilities of 4/(4+3+2), 3/(4+3+2), and 2/(4+3+2). Please refer to lines 3 and 10 in Algorithm 1, which correspond to the insensitive and sensitive cases, respectively, as well as https://anonymous.4open.science/r/Double-Rounding-EF78/code/Super_net/models/roulette_algorithm.py in our code.
> **Lastly**, regarding the absence of the {8, 6, 4, 2} configuration in Table 3, this choice was made to ensure fair comparisons, as most previous work is based on this configuration. Additionally, we have provided results for the {8, 6, 4, 2} configuration below for your reference.
> | Model    | KD | Training | Searching | Fine-tune | Epoch | w8a8  | w6a6  | w4a4  | w2a2  | 5MP   | FP    |
> |:--------:|:---:|:--------:|:---------:|:---------:|:-----:|:-----:|:-----:|:-----:|:-----:|:-----:|:-----:|
> | ResNet18 | ×  | HASB     | ILP       | w/o       | 90    | 70.35 | 70.16 | 70.01 | 64.84 | 70.38 | 69.76 |
> |          | √  | HASB     | ILP       | w/o       | 90    | 70.60 | 70.40 | 70.25 | 65.10 | 70.63 | 69.76 |
> | ResNet50 | ×  | HASB     | ILP       | w/o       | 90    | 76.54 | 76.32 | 75.86 | 72.33 | 76.63 | 76.13 |
> |          | √  | HASB     | ILP       | w/o       | 90    | 76.70 | 76.50 | 76.10 | 72.60 | 76.85 | 76.13 |
>
> **W4:** The accuracy of FP models is inconsistent in experiments.
> **A4:** Thanks for your comments.
> The results in Table 2 are based on the reported numbers from the papers of the compared methods. On one hand, most of these works have not provided open-source code, making it challenging to reproduce their exact results. On the other hand, different works use varying training strategies, which can lead to inconsistencies in FP model accuracy. We have aligned our results with as many state-of-the-art methods as possible to ensure fair comparisons.
> Additionally, while AdaBits shows a smaller relative accuracy drop, it achieves this because it utilizes a greater number of training epochs. We also conduct experiments with increased training epochs and observe that longer training epochs effectively improve the multi-precision results. The details are provided below and the proposed method shows a much better performance than AdaBits under the same training configuration.
> | Model    | Dataset  | Method  | Epoch | w4a4  | w3a3  | w2a2  | FP    |
> |----------|----------|---------|-------|-------|-------|-------|-------|
> | ResNet50 | ImageNet | AdaBits | 150   | 76.10 | 75.80 | 73.20 | 75.00 |
> |          |          | Ours*   | 90    | 76.42 | 75.82 | 73.28 | 76.13 |
> |          |          | Ours*   | 150   | 76.65 | 75.96 | 73.40 | 76.13 |
> |          |          | Ours*   | 300   | 76.78 | 76.05 | 73.50 | 76.13 |
>
> **W5:** It is unclear how knowledge distillation is performed in multi-precision training.
> **A5:** Thanks for your valuable question. We drew inspiration from the progressive in-place distillation of [3], i.e., self-distillation. However, we apply it to the multi-precision quantization method in this paper. Specifically, higher bit-widths distill to their neighboring lower bit-widths, such as 8-bit distills to 6-bit, 6-bit distills to 4-bit, and 4-bit distills to 2-bit.
>
> [3] self-knowledge distillation with progressive refinement of targets. CVPR’2021
>
> Sincerely,
>
> Paper #5733 Authors

---

> ### Author Response · Authors · 2024-11-21
> **Looking forward to your feedback**
>
> Dear Reviewer qsNB,
>
> With the discussion phase nearing the end, we would like to know whether the responses have addressed your concerns.
>
> Should this be the case, we are encouraged that you raise the final rating to reflect this.
>
> If there are any remaining concerns, please let us know. We are more than willing to engage in further discussion and address any remaining concerns to the best of our abilities.
>
> We are looking forward to your reply. Thank you for your efforts in this paper.
>
> Best regards,
>
> Paper #5733 Authors

---

> > ### Comment · Reviewer_qsNB · 2024-11-26
> >
> > Thanks for your reply! The clarification on the memory overhead of the ALRS strategy is helpful.
> >
> > However, I still can't understand why dynamic bit switching can't just occur between 8/6/4/2 bits, and instead must introduce the floating-point type and use a double rounding scheme. Can the proposed method be applied to an INT8 model to perform bit switching?
> >
> > Another question: Since the mixed-precision training is initialized by the multi-precision training, what is the total number of training epochs of both stages in the experiment in Table 3? Is it 90?

---

> ### Author Response · Authors · 2024-11-26
>
> Dear Reviewer qsNB，
>
> Thank you for your constructive feedback and for acknowledging the efforts we have made to address your previous concerns.
>
> Firstly, we would like to clarify that floating-point types are introduced only during the training phase to facilitate quantization to lower bit-widths. However, in actual hardware implementations, only the highest integer type (e.g., INT8) is retained for the model, ensuring that bit-switching occurs strictly between 8/6/4/2 bits **without involving floating-point types in inference**. Additionally, some methods (e.g., MultiQuant[3] and EQ-Net[4]) necessitate retaining a floating-point model for bit-switching during inference, whereas our approach does not.
>
> Secondly, similar to prior one-shot mixed-precision methods (e.g., Bit-Mixer[1], ABN [2], MultiQuant[3]), our mixed-precision training utilizes the multi-precision trained model as the initial model to achieve better convergence for the optimal super-net. Thus, the total number of training epochs across both stages is 180 (90 epochs for multi-precision training followed by 90 epochs for mixed-precision training). Note that other methods also adopt a two-stage approach with a doubled epoch count (e.g., **ours: 180 vs. others: 320**).
>
> Finally, if we directly use a float model as the initialization for mixed-precision training, the joint optimization of models quantized to various bit-widths simultaneously poses significant challenges. This can lead to a slight degradation in the quality of the super-net or even convergence issues, as previously demonstrated by prior works (e.g., Bit-Mixer[1], ABN [2], MultiQuant[3]). We hope this explanation resolves your concerns.
>
> [1] Bit-mixer: Mixed-precision networks with runtime bit-width selection. ICCV'2021.
>
> [2] Arbitrary bit-width network: A joint layer-wise quantization and adaptive inference approach, ACMMM'2022.
>
> [3] Multiquant: Training once for multi-bit quantization of neural networks. IJCAI'2022.
>
> [4] Eq-net: Elastic quantization neural networks. ICCV'2023.
>
> Sincerely,
>
> Paper #5733 Authors

---

### Official Review · Reviewer_VJoM · 2024-11-06

**Soundness:** 2
**Presentation:** 2
**Contribution:** 2
**Rating:** 5
**Confidence:** 5

**Summary:**

This paper targeted two quantization applications, i.e. multi-precision and mixed-precision. Models in former case hold only one set of weights under a given precision (e.g. 8bit) and then allow users to switch the entire model to another precision (e.g. 2bit) while the 2bit weights are converted from the given 8bit weights. A "double rounding" scheme was proposed for this translation of the given weights into the desired lower precision. For mixed-precision case, when deciding the optimal precisions for each individual layer, a Hessian Matrix Trace was employed as an indicator to identify sensitive layers, which will be assigned with higher probability to select higher precisions. The proposed methods were tested on image classification, object detection, and zero-shot commonsense reasoning tasks.

**Strengths:**

1. Good amount of experimental data including a few representative models on vision and NLP tasks.
2. proposed HASB method seems to be effective on mixed-precision case.

**Weaknesses:**

1. Rounding or Flooring?
According to Eq 2 and the name "double rounding", the conversion of INT8 weights to lower precision weights should be a rounding process. However, Line 203 states that "bit-shift" was used to perform the "divide by 2" operation. As bit-shift is not a rounding but a floor division, e.g. 7>>1 = 3 instead of 4, could author please clarify which operation is really used in this method? (One should note that the 4th illustration in Fig. 2 also seems to be a floor operation instead of rounding.)

2. Generalization concerns.
a) ALRS seems to be a purely empirical hyper-parameter tuning.
Comparing Fig 3a and 3b, it seems like gradient magnitude for 4, 6, 8bit are far from clipping bound in Eq. 6, i.e. 1.0. Fig 3c/3d also seems to suggest that gradient clipping is only needed for 2bit case. Roughly speaking, Eq 6 simply suggests to scale LR by 0.1 for every 2 bits reduced in precision. However, Fig 3c/3d and Appendix Fig 7, 8, 9 do not show any apparent differences in gradient magnitude to justify this scaling. This type of empirical rule most likely will not be generalizable when model architecture and dataset changed, as the LLM example in 4.3, author mentioned that ALRS is not applied in that case possibly due to this reason.

b) Threshold for Hessian Matrix Trace
Hessian indicator and HASB seems to work well in Fig. 5a and 5b, but when 2bit is not included, as in Fig. 5c, the improvement is not clear. Author may want to show the HMT distribution and sensitive layers assignment in this case and discuss the differences. On the other hand, the threshold for sensitive layers is the mean HMT of all layers in this work. This kind of relative threshold could be dangerous if all HMT are fairly close, because roughly half of the layers will be labeled as sensitive and encouraged to use higher precision, which will lose the benefit of mixed-precision. Maybe additional checks need to be applied for HASB, for example, limit the total number of sensitive layers.

3. Repeat paragraph, typos, plot being too small.
The very first paragraph in Introduction is repeated at Line 41 to 45, Line 354 "DeepWise" should be "Depth-wise", Fig 3 and 4c/d, fonts are too small and illegible. Author may want to proofread this manuscript one more time.

**Questions:**

please see Weakness

---

> ### Author Response · Authors · 2024-11-19
>
> Dear Reviewer VJoM,
>
> We greatly appreciate your time and efforts for this paper's improvement and here are our responses:
>
> **W1:** Rounding or Flooring?
> **A1:** Thank you for carefully pointing out the mistake in Fig. 2 and the confusion in Line 203.
> **Firstly**, the statement in Line 203 that "bit-shift was used to perform the divide by 2 operation" is indeed misleading. What we actually perform is dividing $\widetilde{w}_h$  by (1≪∆) (equivalent to dividing $\widetilde{w}_h$ by $2^Δ$), followed by a rounding operation. This is reflected in the quantization operator in our training and inference code, specifically at line 145 in https://anonymous.4open.science/r/Double-Rounding-EF78/code/Multi_Precision/models/quan_lsq.py. Regarding your observation about the right-shift operation performing floor division, you are correct, and this led to the incorrect display in the fourth subfigure of Fig. 2. We have updated Fig. 2 in the revised version, and it now clearly shows the correct results of rounding, which provides a more uniform quantization effect and ensures better numerical stability, especially at lower bit-widths. Below is the Python code used to generate the corrected figure. Thank you again for bringing this to our attention.
> ```
> def Double_Round(input, bit_list):
>     scale = 1 << bit_list[-1]-1
>     for bit in bit_list:
>         if bit<bit_list[-1]:
>             diff_bit = bit_list[-1] - bit
>         else:
>             diff_bit = 0
>         if bit==1:
>             Qn, Qp =-1,1
>             q_res = np.clip(np.sign(input), Qn, Qp)
>         else:
>             Qn, Qp =(-1*(1<<bit-1),(1<<bit-1)-1) #-128---127
>             q_res_before = np.round(input * scale)
>             q_res = np.clip(np.round(q_res_before / (1 << diff_bit)), Qn, Qp) # right
>             # q_res = np.clip(np.round(q_res_before >> diff_bit), Qn, Qp) # error
>         d_res = q_res / scale * (1 << diff_bit)
>     return d_res
> input = np.arange(-1, 1.01, 0.000001)
> bit_list = [2, 3, 4]
> ```
> **Secondly**, regarding the hardware implementation of the division by $2^Δ$, this is efficiently achieved by left-shift Δ from 1, avoiding the error of floor division by explicit right-shift operation. We have corrected Line 203 in the revised version to clarify this explanation. We greatly appreciate your thoughtful feedback and hope these updates and explanations address your concerns.
>
> **W2-a:** Generalization concerns?
> **A2-a:** Thank you for your insightful feedback regarding the generalization of the ALRS method.
> **Firstly**, as you noted, without considering outliers, the differences in the gradient magnitude of the quantization scale for 4, 6, and 8 bits are generally not significant compared to 2-bit. This is because the competitive relationship in the one-shot joint optimization for multi-precision primarily arises between higher precisions (8,6,4 bits) and the lowest precision (2-bit). However, we have also conducted experiments without the 2-bit configuration, showing that ALRS still provides some benefits. For instance, MobileNetV2 in Table 6 uses {8,6,4}-bit without 2-bit, and ALRS improve the accuracy with average 2.18%, which further indicate the effectiveness of ALRS.
> **Secondly**, regarding the LLM’s experiments in Section 4.3, we opted not to use ALRS in that case simply to highlight the effectiveness of our proposed Double Rounding method without additional techniques and the results confirm that the proposed method is simple and effective.
> | Model        | Precision | HellaSwag | Obqa  | WinoGrande | ARC-c | ARC-e | boolq | piqa  | Avg. ↑ |
> |:--------:|:---------:|:---------:|:-----:|:----------:|:-----:|:-----:|:-----:|:-----:|:------:|
> | TinyLlama 1.1B      | FP        | 59.20     | 36.00 | 59.12      | 30.12 | 55.25 | 57.83 | 73.29 | 52.99  |
> | iteration-step 8000 | w8a8      | 57.79     | 35.60 | 58.72      | 30.20 | 53.24 | 62.69 | 72.14 | 52.91  |
> |                     | w6a6      | 51.57     | 30.00 | 57.77      | 25.34 | 46.76 | 56.85 | 68.39 | 48.10  |
> |                     | w4a4      | 25.93     | 24.60 | 51.85      | 28.16 | 25.29 | 51.59 | 49.89 | 36.76  |
> |                     | w2a2      | 25.81     | 27.40 | 51.22      | 26.45 | 25.93 | 62.17 | 50.16 | 38.45  |
> | +ALRS               | w8a8      | 57.95     | 35.75 | 58.91      | 30.36 | 53.42 | 63.20 | 72.45 | **53.10**  |
> |                     | w6a6      | 51.70     | 30.25 | 58.05      | 25.50 | 47.02 | 57.01 | 68.55 | 48.30  |
> |                     | w4a4      | 26.05     | 24.80 | 52.10      | 28.32 | 25.48 | 52.01 | 50.03 | 36.93  |
> |                     | w2a2      | 25.90     | 27.55 | 51.38      | 26.60 | 26.15 | 62.41 | 50.35 | 38.60  |
>
> **Overall**, while the competitive relationship in the one-shot joint optimization does exist, we plan to explore more advanced learning rate optimization techniques in the future.

---

> ### Author Response · Authors · 2024-11-19
>
> **W2-b:** Improved threshold for Hessian Matrix Trace Hessian?
> **A2-b:**  Thank you for your constructive feedback.
> Regarding the limited improvement in the {8,4}-bit mixed-precision experiment shown in Figure 5c, there are two possible reasons: (1) Without the 2-bit configuration, the competition among higher precisions (8, 6, 4 bits) is reduced, so using HMT alone to determine sensitivity may not reveal significant differences. (2) It may also be due to the simple thresholding strategy we used for sensitive layers, i.e., setting the threshold as the mean HMT across all layers. As you pointed out, this approach could label approximately half of the layers as sensitive, encouraging higher precision for them, which could reduce the benefits of mixed-precision.
>
> To address this, we have experimented with an alternative approach where we set the threshold based on a top-k strategy, selecting the mean HMT of the top-k most sensitive layers. The results are shown in the following table and as we can see, this approach refines the sensitivity threshold and can further enhance mixed-precision performance. Thanks again for your valuable suggestion.
> | Model    | Top-k | KD | Training | Searching | Fine-tune | Epoch | w4a4  | w3a3  | w2a2  | 3MP   | FP    |
> |:--------:|:-----:|:---:|:--------:|:---------:|:---------:|:-----:|:-----:|:-----:|:-----:|:-----:|:-----:|
> | ResNet18 | w/o   | ×  | HASB     | ILP       | w/o       | 90    | 69.80 | 68.63 | 64.88 | 68.85 | 69.76 |
> |          | k=5   | ×  | HASB     | ILP       | w/o       | 90    | 70.10 | 68.90 | 65.05 | 69.00 | 69.76 |
> |          | k=10  | ×  | HASB     | ILP       | w/o       | 90    | **70.35** | **69.20** | 65.23 | **69.30** | 69.76 |
> |          | k=15  | ×  | HASB     | ILP       | w/o       | 90    | 69.85 | 68.70 | **65.50** | 68.90 | 69.76 |
>
> **W3:** Repeat paragraph, typos, plot being too small.
> **A3:** Thank you for your valuable feedback. We have carefully addressed the issues you mentioned, including the repeated paragraph, typos (e.g., "Depth-wise" instead of "DeepWise"), and the font size in Fig 3 and 4c/d. We have corrected these errors in the revised version and thoroughly proofread the manuscript for other potential typos. Thanks again for your attentive review.
>
> Sincerely,
>
> Paper #5733 Authors

---

> ### Author Response · Authors · 2024-11-21
> **Looking forward to your feedback**
>
> Dear Reviewer VJoM,
>
> With the discussion phase nearing the end, we would like to know whether the responses have addressed your concerns.
>
> Should this be the case, we are encouraged that you raise the final rating to reflect this.
>
> If there are any remaining concerns, please let us know. We are more than willing to engage in further discussion and address any remaining concerns to the best of our abilities.
>
> We are looking forward to your reply. Thank you for your efforts in this paper.
>
> Best regards,
>
> Paper #5733 Authors

---

> > ### Comment · Reviewer_VJoM · 2024-11-25
> >
> > Thanks to the author for providing additional experimental data. However, the difference between with and without ALRS in A2-a is still quite small. A2-b using top-k seems to work well in terms of accuracy, but it would be nice to include the observed benefit of using this method, such as Table 8 and Fig 5 in the original manuscript. In other words, user may wonder how to determine k in this case?
> >
> > Overall, the quality of the work is mildly improved after the new revision, but I'd like to keep my evaluation unchanged.

---

> ### Author Response · Authors · 2024-11-26
>
> Dear Reviewer VJoM,
>
> Thank you for your constructive feedback and for acknowledging the efforts we have made to address your previous concerns.
>
> Firstly, the small improvement of ALRS in A2-a could be due to the discrepancy in optimal hyperparameters between NLP and CV tasks and **we are the first to conduct experiments on LLMs for multi-precision quantization** to the best of our knowledge. Due to the time limitation, we cannot tune the hyperparameters for this time and we will include the updated experimental results in the final manuscript. Moreover, the results of MobileNetV2 also demonstrate the effectiveness of the proposed method while **the previous methods cannot converge for MobileNetV2 as stated in [1][2][3][4]**.
>
> Secondly, thanks for inspiring us for the Top-K strategy. We have updated Fig. 5 (b) in the revised version to include results with the Top-k strategy applied. Regarding the selection of the k value for the Top-k strategy, our experiments suggest that 10 is an empirically ideal choice. It is noted that the Top-k strategy is unrelated to Table 8, as Table 8 represents results for multi-precision experiments.
>
> Finally, if you feel that your further concerns are addressed, we would appreciate updating your evaluation to reflect that. Thanks very much for your time and efforts for this paper. Looking forward to your reply.
>
> [1] Bit-mixer: Mixed-precision networks with runtime bit-width selection. ICCV'2021.
>
> [2] Arbitrary bit-width network: A joint layer-wise quantization and adaptive inference approach, ACMMM'2022.
>
> [3] Multiquant: Training once for multi-bit quantization of neural networks. IJCAI'2022.
>
> [4] Eq-net: Elastic quantization neural networks. ICCV'2023.
>
> Sincerely,
>
> Paper #5733 Authors

---

> > ### Author Response · Authors · 2024-12-02
> > **Looking forward to your feedback**
> >
> > Dear Reviewer VJoM,
> >
> > Sorry to interrupt you again and we have updated the responses to your further concerns. With the discussion phase nearing the end (Dec.2nd AoE), we would like to know whether the responses have addressed your concerns. Should this be the case, we are encouraged that you raise the final rating to reflect this.
> >
> > If there are any remaining concerns, please let us know. We are more than willing to engage in further discussion and address any remaining concerns to the best of our abilities. We are looking forward to your reply. Thanks very much for your time and efforts on this paper.
> >
> > Best regards,
> >
> > Paper #5733 Authors

---

### Author Response · Authors · 2024-11-25
**Looking forward to your reply**

Dear reviewers,

With the discussion phase nearing the end, we would like to know whether the responses have addressed your concerns.

Should this be the case, we are encouraged that you raise the final rating to reflect this.

If there are any remaining concerns, please let us know. We are more than willing to engage in further discussion and address any remaining concerns to the best of our abilities.

We are looking forward to your reply. Thank you for your time and efforts in this paper.

Best regards,

Paper #5733 Authors

---

### Note · Authors · 2025-01-22

I have read and agree with the venue's withdrawal policy on behalf of myself and my co-authors.